# Improved WOA and its application in feature selection

**Wei Liu[1], Zhiqing Guo[1]\*, Feng Jiang[1], Guangwei Liu[2], Dong Wang[2], Zishun Ni[1]**

**1** College of Science, Liaoning Technical University, Fuxin, Liaoning, China, **2** College of Mines, Liaoning Technical University, Fuxin, Liaoning, China

\* mathgzq@gmail.com

## Abstract

Feature selection (FS) can eliminate many redundant, irrelevant, and noisy features in high-dimensional data to improve machine learning or data mining models' prediction, classification, and computational performance. We proposed an improved whale optimization algorithm (IWOA) and improved k-nearest neighbors (IKNN) classifier approaches for feature selection (IWOAIKFS). Firstly, WOA is improved by using chaotic elite reverse individual, probability selection of skew distribution, nonlinear adjustment of control parameters and position correction strategy to enhance the search performance of the algorithm for feature subsets. Secondly, the sample similarity measurement criterion and weighted voting criterion based on the simulated annealing algorithm to solve the weight matrix M are proposed to improve the KNN classifier and improve the evaluation performance of the algorithm on feature subsets. The experimental results show: IWOA not only has better optimization performance when solving benchmark functions of different dimensions, but also when used with IKNN for feature selection, IWOAIKFS has better classification and robustness.

## 1. Introduction

With the continuous development and progress of science and technology and its continuous use in biomedicine, astronomy, agriculture, finance, and engineering, various forms of data have shown exponential growth [1]. However, original datasets usually contain many redundant, uncorrelated, and noisy features, making data mining very difficult [2]. Feature selection (FS) refers to a method that selects the optimal feature subset from many original features and restores all the features in the original datasets as much as possible with the smallest number of features [3]. FS has become a key preprocessing step for machine learning and pattern recognition [4].

According to different evaluation methods, FS was roughly divided into 3 categories: wrapper, filter, and embedding [5]. The wrapper-based model combined the FS process with predetermined learning algorithms (e.g., classifiers) and used learning algorithms to evaluate each feature subset. Its characteristic is that the solution accuracy is high, but the solution speed is slow. The filter-based model evaluated and filtered candidate subsets through the intrinsic attributes of the features without relying on any learning algorithm. Its characteristic is that

**Funding:** This study was funded by the National Natural Science Foundation of China [51974144; 51874160], the Project of Liaoning Provincial

Department of Education [LJKZ0340], and the Discipline Innovation Team of Liaoning Technical University [LNTU20TD-01; LNTU20TD-07] in the form of funds to WL. This study was also supported by the Jie Bang Gua Shuai'(Take The Lead) of Key Scientific and Technological Project For Liaoning Province in the form of funds to GL [2021JH1/10400011].

**Competing interests:** The authors have declared that no competing interests exist.

the solution is fast, but it cannot reflect the relevance of various dimensions when extracting feature subsets. The embedding-based model is a combination of wrapper and filter. The FS mechanism is integrated into the training process of the learning algorithm and features are automatically selected while training the model [6–9].

Searching for feature subsets is a key problem of FS [10]. The process can be regarded as a combinatorial optimization problem to seek the optimal feature subset in a limited feature space. This process can be searched and solved by exhaustive methods and heuristic methods. However, exhaustive and heuristic methods need to traverse every dataset sample when performing FS, the algorithm search space increases, resulting in increased algorithm time complexity and high computational cost [11].

In order to improve the speed and efficiency of algorithm searching for feature subsets, a class of meta-heuristic algorithms inspired by natural evolution has been proposed. Due to its simple heuristic mechanism and strong global exploration ability, it has been widely used in various fields. Therefore, researchers use meta-heuristic algorithm as a search strategy for feature subsets in FS, and a series of more meaningful research results have been achieved so far [12–17]. For example: grey wolf optimizer (GWO) [18] was improved by Seth, JK, etc. for binary gray wolf optimization algorithm for intrusion detection [19], and Two binary gray wolf optimization algorithms proposed by Emary E et al. after improving GWO [20], Al-Tashi Q et al. mixed GWO and PSO binary gray wolf algorithm [21]; Too J et al. improved binary ASO algorithm based on S-shaped and V-shaped transfer functions [22] and quadratic binary Harris hawk optimization (QBHHO) algorithm [23]; Kumar L et al. proposed a hybrid binary particle swarm algorithm and sine cosine algorithm (HPSOSCA) using V-shaped transfer function to change particle position [24]; Improved MBO for feature selection problems by Alweshah M et al. [25]; Mafarja M and Mirjalili S improved WOA for feature selection problems [26] and so on. In addition, Remora optimization algorithm (ROA) [27], African vulture optimization algorithm (AVOA) [28], Gorilla troops optimizer (GTO) [29], Wild horse optimizer (WHO) [30], Binary chimp optimization algorithm (BChOA) [31], Arithmetic optimization algorithm (AOA) [32], Aquila optimizer (AO) [33] and other meta-heuristic algorithms are also being explored for feature selection problems.

Among them, WOA is a new meta-heuristic algorithm inspired by the social behavior of humpback whales by Mirjalili et al. [34] to simulate the hunting behavior of whales. Because of its fast convergence speed, high accuracy, and few parameters when solving some optimization problems, WOA is widely used in various engineering practices and FS. According to the different improvement mechanisms, the existing WOA-based FS methods are roughly divided into three categories:

The first category is the classic WOA FS method based on binary variants. The classic WOA method is transformed into a binary WOA algorithm through a sigmoid function or a V-shaped function, and the optimal feature subset is searched for using the strong global search ability of WOA, thereby improving the classification accuracy of the dataset. This kind of method is applied in medical dataset [35, 36], breast cancer dataset [37], network intrusion detection [38, 39], spam filtering [40], dimensionality reduction of high dimensional data [41, 42], etc.

The second category is the improved WOA algorithm that enhances the algorithm's global exploration and local development functions. The performance of WOA is enhanced by modifying the parameters and introducing operators, and the efficiency of FS is improved. This type of method usually enhances the exploration and development capabilities of WOA by increasing the diversity of the initial population (Elite Opposition-Based Learning [43] and Chaos strategy [44]), nonlinear correction of control parameters [26, 45, 46] and algorithm location update [47–51] improve the algorithm's search performance for feature subsets.

The third category is the improved strategy of the cross-fusion of the WOA algorithm and other algorithms. The optimization performance of WOA is enhanced by fusing the search characteristics of different algorithms, thus improving the search efficiency of the optimal feature subset. For example, the fusion of different algorithms such as WOA and salp swarm algorithm (SSA) [52], WOA and flower pollination algorithm (FPA) [53], WOA and GWO [54], WOA and simulated annealing (SA) [16], WOA and GA [55] can improve the convergence performance of the original algorithm effectively and achieve meaningful results in FS.

The evaluation of feature subsets is also a key issue of FS [10]. This process can be seen as a binary classification problem to evaluate the optimal feature subset through the classifier. After the meta-heuristic algorithm is used to search the feature subset, the k-nearest neighbors (KNN) classifier and support vector machine (SVM) classifier are usually used to evaluate the feature subset. The KNN classifier is the most used classifier when obtaining the best feature subset on the UCI knowledge base. SVM classifiers are mainly used in different applications such as medical diagnosis, pattern recognition, and image analysis [8, 17]. Studies have shown that different classifiers have significant differences in the results of FS [56]. Improving the KNN classifier and applying it to FS by fusing it with improved WOA is another motivation for the work of this paper.

The above studies have done good work on the FS problem in different periods and different fields. However, no optimization algorithm can completely solve all problems according to NO-Free-Lunch (NFL) [57]. This is also the basis and motivation of the work of this article. Therefore, for two key issues in FS: the search and evaluation of feature subsets. The main contributions of this paper are as follows:

- An improved whale optimization algorithm (IWOA) is proposed based on chaotic reverse elite individuals and skew distribution. IWOA and 8 meta-heuristic algorithms (ASO, GWO, HHO, MFO, MVO, SSA, TSA, and WOA) were compared in two dimensions (30D and 100D) of 8 benchmark functions, verifying that IWOA has good superior performance.

- An improved KNN classifier (IKNN) is proposed based on a weight matrix **M** and a weighted classification strategy. The comparison of IKNN and 5 classifiers (KNN, Naive Bayes, C4.5, SVM, and BP neural network) on 8 datasets have shown that IKNN has good classification performance.

- A FS method based on IWOA and IKNN (IWOAIKFS) is proposed. As a wrapper-based model, IWOAIKFS is an FS method that uses IWOA fusion IKNN to search and evaluate feature subsets.

- IWOAIKFS was applied to evaluate 15 datasets. The results are compared with 15 datasets evaluated with 6 FS methods based on meta-heuristic algorithms (ASO, GWO, HHO, SCA, SSA, and WOA). The comparison has shown that IWOAIKFS has higher classification accuracy and stable performance than 6 FS methods.

The remaining paper is organized as follows: Section 2 is the standard WOA algorithm and an improved whale optimization algorithm (IWOA). In Section 3, the standard KNN classifier is discussed, and the improved KNN classification algorithm (IKNN) is discussed and proposed. In Section 4, an IKNN FS method is discussed and proposed based on IWOA. In Section 5, the experimental results of three experiments are discussed and analyzed. Section 5.3 is the IWOA comparative experiment and discussion. Section 5.4 is the IKNN comparative experiment and discussion. Section 5.5 is the IWOAIKFS comparative experiment and discussion. In Section 6, the current work and possibly future research work are summarized.

## 2. Improved whale optimization algorithm based on chaotic elite reverse individual and skew distribution

### 2.1. WOA

The whale optimization algorithm is a meta-heuristic algorithm inspired by the social behavior of humpback whales by Mirjalili et al. [34] to simulate whales searching for prey, surrounding prey, and a spiral bubble net preying on prey. It is supposed that there are $N$ whales in the population foraging in $d$ dimensional space, the position of the $i^{th}$ whale is $\overrightarrow{X}_i$.

**2.1.1. Searching for prey.** Before determining the approximate location of the prey whales update the location of the group through random walks and location sharing mechanisms. The process can be expressed as:

$$\overrightarrow{X}(t+1) = \overrightarrow{X_{rand}} - \overrightarrow{A} \times \overrightarrow{D_{rand}} \tag{1}$$

$$\overrightarrow{D_{rand}} = |\overrightarrow{C} \times \overrightarrow{X_{rand}}(t) - \overrightarrow{X}(t) \tag{2}$$

Where $t$ represents the current number of iterations, $\overrightarrow{X}(t+1)$ represents the location of the whale at the $(t+1)^{th}$ time, $\overrightarrow{X_{rand}}$ represents the location of the random individual in the whale group (initial group), $\overrightarrow{D_{rand}}$ represents the distance between the current individual and the random individual whale. Besides, $\overrightarrow{A}$ and $\overrightarrow{C}$ are the coefficient vector, which can be expressed as:

$$\overrightarrow{A} = 2a\overrightarrow{r} - a \tag{3}$$

$$\overrightarrow{C} = 2\overrightarrow{r} \tag{4}$$

where $\overrightarrow{r} \in [0, 1]$, $a$ is the algorithm convergence factor, which decreases from 2 to 0 as the iteration progresses, expressed as:

$$a = 2 - \frac{2 \times t}{t_{max}} \tag{5}$$

where $t$ represents the current number of iterations, $t_{max}$ is the maximum iterations of the algorithm. It can be seen from Eqs (1) (3) and (5) that when $\left|\overrightarrow{A}\right| > 1$ the algorithm simulates the process of whale searching for prey and explores the location of the optimal solution in the solution space. As the algorithm continues to iterate, $\left|\overrightarrow{A}\right|$ decreases linearly with $a$, until $\left|\overrightarrow{A}\right| \leq 1$, the algorithm enters the stage of encircling the prey.

**2.1.2. Approaching and encircling the prey.** As the number of iterations increases, the location of the prey is determined by the optimal whale in the whale group. Other whale individuals gradually approach their prey by shrinking and enclosing through position sharing. This process can be expressed as:

$$\overrightarrow{X}(t+1) = \overrightarrow{X_*}(t) - \overrightarrow{A} \times \overrightarrow{D} \tag{6}$$

where $\overrightarrow{X_*}(t)$ represents the optimal solution position vector in the current whale population, $\overrightarrow{D} = |\overrightarrow{C} \times \overrightarrow{X_*}(t) - \overrightarrow{X}(t)|$ represents the distance between the optimal whale individual and other individuals.

**2.1.3. Encircling and capturing prey.** When a school of whales approaches their prey, humpback whales' prey on the prey through the spiral position update method. The spiral position update method is expressed as:

$$\overrightarrow{X}(t+1) = \overrightarrow{D\prime} \times e^{bl} \times \cos(2\pi l) + \overrightarrow{X_*} \tag{7}$$

where $\overrightarrow{D\prime} = \left| \overrightarrow{X_*}(t) - \overrightarrow{X}(t) \right|$ represents the current distance between the whale and its prey, $b$ is a constant defining the shape of the logarithmic spiral. l is a random number between [–1,1]. Assuming that the probability of a whale performing any of these behaviors is 50%, the whale encircling and hunting behavior can be expressed as:

$$\overrightarrow{X}(t+1) = \begin{cases} \overrightarrow{X_*}(t) - \overrightarrow{A} \times \overrightarrow{D}, & p < 0.5 \\ \overrightarrow{D\prime} \times e^{bl} \times \cos(2\pi l) + \overrightarrow{X_*}(t), & p \geq 0.5 \end{cases} \tag{8}$$

where $p$ is a random number between [0, 1].

## 2.2. IWOA

In order to better improve the convergence speed and accuracy of WOA, WOA was improved from 4 aspects: population initialization, probability selection, Parameter correction and position update.

**2.2.1. Population initialization based on chaotic reverse elite individuals.** In the whale optimization algorithm, the initial position of the whale group has great constraints on prey. When the distance between the whale group and the prey is closer, the whale can prey on the prey faster. Therefore, the location of the initial population of whales is important for solving the optimal value of the algorithm. Chaos strategy [58] elite opposition-based learning [59] as effective improvement strategies for population initialization are widely used in generating initialization populations of various meta-heuristic algorithms. However, using chaos strategy alone or using elite opposition-based learning alone to generate the initial population of the algorithm ignores the uniform distribution of the initial population in the solution space and the retention of elite individuals. Therefore, this paper proposed a population initialization strategy based on the chaotic reverse elite individuals by combining the Gaussian chaos strategy and the Elite Opposition-Based Learning.

Answering the question: The more evenly distributed the initial population is in the solution space, the greater the probability that the algorithm finds the optimal value. Compared with random search strategy, chaotic search is widely used in the generation of initial population due to its randomness, ergodicity, non-repetition and other characteristics. However, different chaotic maps have different effects on the initial population of the algorithm. Therefore, in this paper, through the analysis and comparison of Rand, Gauss map, Tent map and Chebyshev map, the chaotic map suitable for the whale optimization algorithm is selected. The initial population and the original initial population generated by the three chaotic maps are shown in Fig 1.

In Fig 1, Fig 1(a)–1(d) represent the original initial whale population, the initial whale population generated by Tent mapping, the initial whale population generated by Chebyshev mapping, and the initial whale population generated by Gauss Generated initial whale population. As can be seen from Fig 1, from the point of view of the generation of the initial whale population, the whale population generated by Gauss map is more evenly distributed in space, which provides a better guarantee for the global optimization of the algorithm.

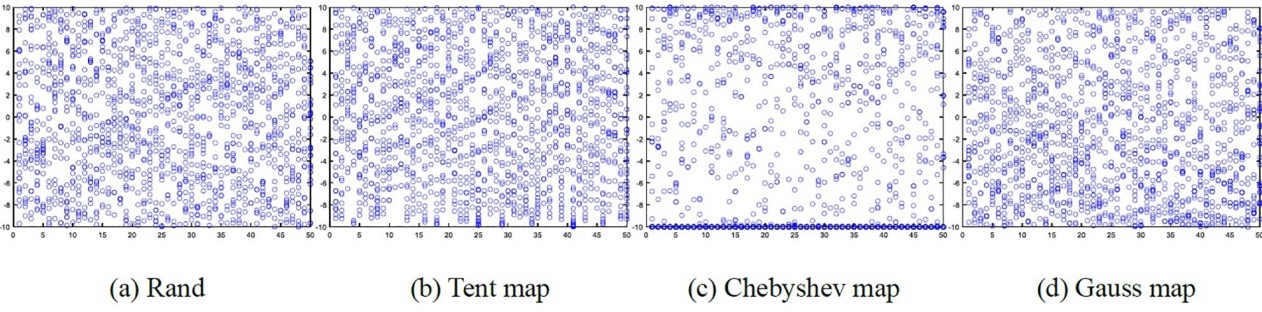

**Fig 1. Initial population generated by 3 kinds of maps in 1000 iterations.** (a) Rand, (b) Tent map, (c) Chebyshev map, (d) Gauss map.

Gauss/mouse map:

$$x_{k+1} = \begin{cases} 1, & x_k = 0 \\ \dfrac{1}{\mod(x_k, 1)}, & otherwise \end{cases} \tag{9}$$

Where $x_k$ is the $k^{th}$ chaos number, $k$ is the number of iterations, $x \in (0, 1)$.

Elite Opposition-Based Learning [59, 60]:

Definition 1 (opposite solution) supposes that a feasible solution of the current population in the $d$ dimensional search space is $\overrightarrow{X} = (x_1, x_2, \cdots, x_d)$ ($x_j \in [a_j, b_j]$), then its opposite solution is $\overrightarrow{\underline{X}} = (\underline{x}_1, \underline{x}_2, \cdots, \underline{x}_d)$, where $\underline{x}_j = r(a_j + b_j) - x_j, r \in rand[0, 1]$.

Definition 2 (elite opposite solution) supposes that extreme points of ordinary individuals in the current population by the elite individuals in the population, that is

$$X_{i,j}^E = \left( X_{i,1}^E, X_{i,2}^E, \cdots, X_{i,d}^E \right) \tag{10}$$

Where $i = 1, 2, \cdots, N, j = 1, 2, \cdots, d, X_{i,j}^E \in [lb_j, ub_j], lb_j = \min(\overrightarrow{X}_{i,j}), ub_j = \max(\overrightarrow{X}_{i,j}), lb_j$ and $ub_j$ are the lower and upper bounds of the dynamic boundary, and the opposite solutions $\overline{X_{i,j}^E} = \left( \overline{X_{i,1}^E}, \overline{X_{i,2}^E}, \cdots, \overline{X_{i,d}^E} \right)$ can be defined as:

$$\overline{X_{i,j}^E} = r \times (lb_j + ub_j) - X_{i,j}^E \tag{11}$$

Where $r \in rand[0, 1]$. If $\overline{X_{i,j}^E}$ exceeds the boundary, then set

$$\overline{X_{i,j}^E} = rand(lb_j + ub_j) \tag{12}$$

According to Gauss/mouse map and Elite Opposition-Based Learning, the initialization method of chaos reverses the elite individual population proposed in this paper is as follows.

Algorithm 1. Chaotic elite reverse individual.

```
Input: N, d, lb = lb_j, ub = ub_j
1: Initialize the Positions1 with Gauss/mouse map (Eq 9)
2: Initialize the Positions2 with Elite Opposition-Based Learning
3: for1 i = 1:N
4:   for2 j = 1:d
5:     fitness1(i,j) = fobj(Positions1(i,j))
```

```
 6:      fitness2(i,j) = fobj(Positions1(i,j))
 7:      if1 fitness1(i,j)<fitmess2(i,j)
 8:        Positions1(i,j) = Positions2(i,j)
 9:      end if1
10:      if2 Positions1(i,j)<lb
11:        Positions1(i,j) = rand()*(ub-lb)+lb
12:      end if2
13:      if3 Positions1(i,j)>ub
14:        Positions1(i,j) = rand()*(ub-lb)+lb
15:      end if3
16:    end for2
17: end for1
18: X⃗_rand = Positions1
Return: X⃗_rand
```

**2.2.2. Probability selection strategy based on the skew distribution.** WOA assumes that the probability of a group of whales choosing to surround and prey is 50%, and the probability $p$ generated by each iteration of the algorithm obeys a uniformly distributed random number between [0,1], which is inconsistent with the animal hunting guidelines in the actual nature. In nature, when a predator finds a prey, the probability of surrounding and preying on the prey changes accordingly with time. The probability of its generation does not obey a uniform distribution.

In order to improve the global exploration ability and convergence accuracy of WOA, a new probability generation method is proposed. This method divides the WOA iteration process into three periods and then corrects the probabilities generated in each period.

1. Early iteration. When an individual whale finds prey, the remaining whales quickly move close to the optimal individual through the position sharing mechanism. At this time, the probability of surrounding the prey is greater than 0.5. Therefore, the probability of hunting behavior in the early stage of the algorithm iteration follows a negative skew distribution of 0.8.

2. Mid iteration. The whales that are close to the prey have already surrounded the prey, and the groups of whales that are far away are constantly approaching the prey. At this time, the probability of the entire group of whales choosing to surround and prey is equal. Therefore, the whale encircling and hunting behavior at this time is given the probability obeys the uniform distribution between [0, 1].

3. Late iteration. Assuming that the group of whales has surrounded the prey and began to attack the prey. However, the probability of the whale's successful predation follows a distribution of less than 0.5 due to the prey's own desire to survive. Therefore, the probability of whale encircling and hunting behavior in the late iteration of the algorithm follows 0.2 Positive skew distribution.

According to the above description, the probability generation equation is given as:

$$p = \begin{cases} rsn(1, 0.75, 0.5, 0.1), & t \le t_{\max}/4 \\ rand(), & t_{\max}/4 < t < t_{\max}/2 \\ rsn(1, 0.25, 0.5, 0.1), & t \ge t_{\max}/2 \end{cases} \tag{13}$$

Where *rsn*(*n,location,scale,shape*) is the skew distribution random number generation method proposed by Azzalini A [61], $t_{\max}$ represents the maximum iterations of the algorithm. Fig 2 shows the probability generation method of whale social behavior in three different periods.

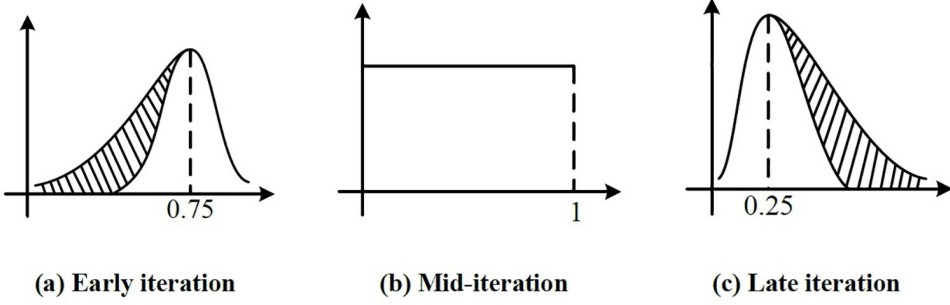

**(a) Early iteration**       **(b) Mid-iteration**       **(c) Late iteration**

**Fig 2. Probability generation of whale social behavior.** (a) Early iteration, b) Mid-iteration, c) Late iteration.

Fig 2 and Eq (13) show that the generated probability range is not between [0, 1]. Therefore, the boundary of the generated probability is constrained, namely

$$
p = \begin{cases} 1, \, p > 1 \\ 0, \, p < 1 \end{cases} \tag{14}
$$

**2.2.3. Nonlinear correction strategy of $a$ and $\overrightarrow{C}$.** In WOA, $\overrightarrow{A}$ and $\overrightarrow{C}$ are important parameters that control whales to explore, surround and prey on prey. The value of $\overrightarrow{A}$ is determined by the convergence factor $a$. Fig 3 shows the changes of WOA's original parameters $a$ and $\overrightarrow{C}$ under 1000 iterations. The convergence factor $a$ shows a linear downward trend, indicating that the distance between the whale and the prey shows a linear downward trend. $\overrightarrow{C}$ is a uniformly distributed random number between 0 and 2, indicating that the distance between the whale and the prey changes randomly. $\overrightarrow{C}$ has no obvious effect on the algorithm's global exploration and local mining.

Therefore, the parameters $a$ and $\overrightarrow{C}$ are modified to speed up the convergence speed of WOA. The modified convergence factor $a$ and $\overrightarrow{C}$ are:

$$
a = \begin{cases} \dfrac{1}{1 + \exp\left(\dfrac{(t - 0.25 t_{\max})}{0.025 t_{\max}}\right)} + 1, t \leq \dfrac{t_{\max}}{2} \\[3mm] 1 - \dfrac{2t}{t_{\max}}, \quad t > \dfrac{t_{\max}}{2} \end{cases} \tag{15}
$$

$$
\overrightarrow{C} = 2rsn(1, 1, 0.7, 0.1) \tag{16}
$$

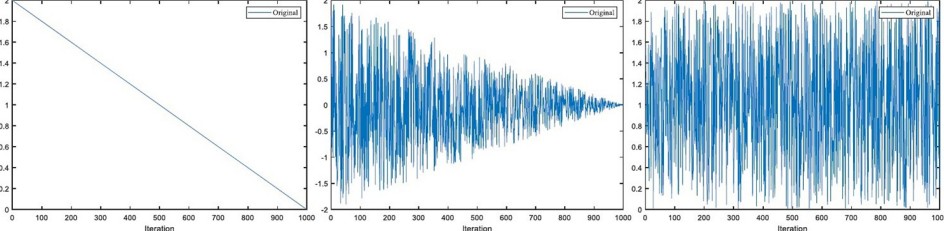

**Fig 3. Original parameter $a$-value, $\overrightarrow{A}$-value and $\overrightarrow{C}$-value under 1000 iterations.**

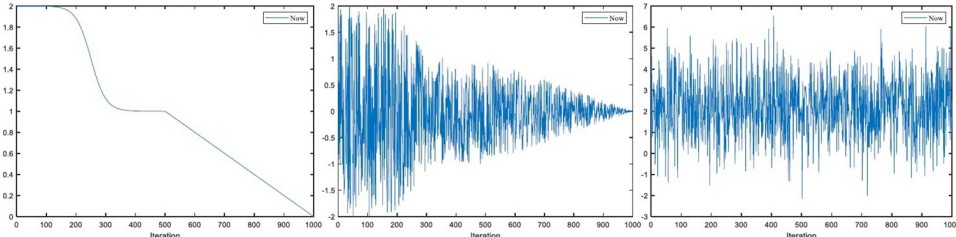

**Fig 4. Modified parameter $a$-value, $\overrightarrow{A}$-value and $\overrightarrow{C}$-value under 1000 iterations.**

Where $a \in [2, 0]$, $t$ is the current number of iterations, and $t_{\max}$ is the maximum iterations of the algorithm. The iterative curve of the updated parameters $a$, $\overrightarrow{A}$ and $\overrightarrow{C}$ at 1000 times is shown in Fig 4.

**2.2.4. Location update strategy.** In WOA, the whale group approaches the optimal whale individual and surrounds the prey when $\left|\overrightarrow{A}\right| < 1$. In order to speed up the process of whales moving to the optimal individual whale and quickly encircle their prey; a nonlinear decreasing disturbance factor is introduced to enhance the local mining capability of the algorithm and improve the accuracy of the algorithm's convergence.

$$\overrightarrow{X_*}(t) - \overrightarrow{A} \times \overrightarrow{D} \times \omega, \quad \left|\overrightarrow{A}\right| < 1 \tag{17}$$

where $\overrightarrow{X_*}(t)$ represents the optimal solution position vector in the current whale population, $\overrightarrow{A}$ and $\overrightarrow{C}$ are coefficient vectors, $\omega$ is a nonlinear decreasing perturbation factor, which is defined as:

$$\omega = 0.8 \times \cos(\frac{\pi}{2t_{\max}} \times t) + 0.2 \tag{18}$$

The revised position update equation is:

$$\overrightarrow{X}(t+1) = \begin{cases} \begin{cases} \overrightarrow{X_*}(t) - \overrightarrow{A} \times \overrightarrow{D} \times \omega, |\overrightarrow{A}| < 1 \\ \left(\overrightarrow{X_*}(t) - \overrightarrow{A} \times \overrightarrow{D}\right), \ |\overrightarrow{A}| \geq 1 \end{cases}, \ p < 0.5 \\ \overrightarrow{D}\prime \times e^{bl} \times \cos(2\pi l) + \overrightarrow{X_*}(t), \ p \geq 0.5 \end{cases} \tag{19}$$

where $\overrightarrow{D} = \overrightarrow{C} \times \overrightarrow{X_*}(t) - \overrightarrow{X}(t)$ represents the distance between the optimal whale individual and other individuals and $p$ is the probability generated by Eqs (13) and (14).

## 2.3. Pseudo-code of the IWOA algorithm

In summary, an improved whale optimization algorithm (IWOA) executes pseudocodes as shown in Algorithm 2.

Algorithm 2. Improved whale optimization algorithm (IWOA).

```
Input: N = Total populations, d, tmax
1: Initialize the whales population Xi(i = 1, 2, ···, N) with Algorithm
1
2: Calculate the fitness of each search agent
3: X* = the best search agent
4: while (t<tmax)
```

```
 5:   for each search agent
 6:     Update a (Eq 15), A, C (Eq 16), l
 7:     Calculate p (Eqs 13 and 14)
 8:     if1 (p<0.5)
 9:       if2 (|A|< 1)
10:         Update the position of current search agent (Eq (17))
11:       else if2 (|A| ≥ 1)
12:         Select a random search agent (X_rand)
13:         Update the position of current search agent (Eq (1))
14:       end if2
15:     else if1 (p ≥ 0.5)
16:         Update the position of current search agent (Eq (19))
17:     end if1
18:   end for
19:   Check the boundary and amend it
20:   Calculate the fitness of each search agent
21:   Update X* if there is a better solution
22:   t = t+1
23: end while
Return: X*
```

## 2.4. Time complexity analysis of IWOA

1. The initialization of the population process needs $\mathcal{O}(N \times d)$ time, where $N$ is the population size, and $d$ defines the dimension of a given test problem.

2. Calculate the $a$ and $p$ needs $\mathcal{O}(t_{max})$, where $t_{max}$ is the maximum number of iterations.

3. Calculate the fitness of each search agent needs $\mathcal{O}(t_{max} \times N \times d + t_{max})$ time.

Hence, the total time complexity of IWOA algorithm is $\mathcal{O}(t_{max} \times N \times d + t_{max})$.

# 3. IKNN based on M and weighted classification strategy

## 3.1. KNN

KNN [62] is a supervised classification algorithm proposed by COVER and HART. KNN is widely used in various fields due to its simple and intuitive idea. The basic principles of KNN classification are:

1. Express the test sample as a feature vector consistent with the training sample set.

2. Calculate the distance between the test sample and each training sample according to the distance function, and select the K samples with the smallest distance from the test sample as the KNN of the test sample.

3. According to the principle of "majority voting", the class with the most occurrences among the KNN are selected as the test sample class.

$k$-value, distance function and "voting method" are important parameter criteria for KNN algorithm classification. The $k$-value represents the number of reference samples selected, which is determined by the actual problem requirements. The distance function corresponds to a non-negative function, used to describe the Similarity between different samples. The distance function can be defined as:

$$D(\overrightarrow{x}, \overrightarrow{y}) = \sqrt{(x_i - y_i)^{\mathsf{T}} \mathbf{I}(x_i - y_i)} \tag{20}$$

Where $D(\overrightarrow{x}, \overrightarrow{y})$ represents the distance between the training sample set $\overrightarrow{x}$ and the test sample set $\overrightarrow{y}$, $x_i$ is the $i^{th}$ attribute of the training sample set $\overrightarrow{x}$, $y_i$ is the $i^{th}$ attribute of the test sample set $\overrightarrow{y}$, $i = 1, \cdots, m$, $m$ is the feature dimension, $I \in \mathbb{R}^n$ is the distance measurement matrix (identity matrix). Assuming that there are $J$ classes in $y_i$'s KNN, the majority voting is:

$$Vot(y_i, C_j) = \max\left\{\sum_{i=1}^{K} Pa(a_i, C_j)\right\} \tag{21}$$

Where $Vot(y_i, C_j)$ indicates that the test sample $y_i$ is the number of class $C_j$, $Pa(a_i, C_j)$ indicates whether the $a_i$ sample among the KNN of $y_i$ belongs to the class $C_j$, which is defined as:

$$Pa(a_i, C_j) = \begin{cases} 1, & a_i \in C_j \\ 0, & a_i \notin C_j \end{cases} \tag{22}$$

Where $j = 1, \cdots, J$.

### 3.2. IKNN

**3.2.1. Sample similarity measurement criterion based on M-matrix.** When classifying complex datasets, the classic KNN algorithm used simple Euclidean distance as the measure of similarity between samples and assigned the same weight to the sample distance through the identity matrix **I**, leading to an exaggeration of weak attributes and weakening of strong attributes easily, thus leading to inaccurate classification.

Based on the simulated annealing algorithm [63], the weight matrix **M** is constructed to replace the identity matrix **I**, then the Eq (20) can be modified as:

$$D(\overrightarrow{x}, \overrightarrow{y}) = \sqrt{(x_i - y_i)^\mathsf{T} \mathbf{M}(x_i - y_i)} \tag{23}$$

where the weight matrix **M** is based on the simulated annealing algorithm. For the datasets under study, the value of the distance weight matrix **M** is adaptively generated with the iteration of the algorithm. Therefore, the **M** is:

$$\begin{cases} D^*(\overrightarrow{x}, \overrightarrow{y}) = 0, & l_i \in C_j \\ D^*(\overrightarrow{x}, \overrightarrow{y}) \geq 1, & l_i \notin C_j \end{cases} \tag{24}$$

Where $l_i$ is the label of the test sample $x_i$ and $C_j$ is the $j^{th}$ class. The objective function is set:

$$\min Z(\mathbf{M}) = \sum_{i=1}^{n} \sum_{j=1}^{n} \left(D(\overrightarrow{x}, \overrightarrow{y}) - D^*(\overrightarrow{x}, \overrightarrow{y})\right)^2 \tag{25}$$

The simulated annealing algorithm is used to solve the weight matrix **M**, and the pseudo code is executed as shown in Algorithm 3.

**3.2.2. Weighted voting criteria.** KNN adopted the "majority voting" to identify test samples. The selected sample was tested as the center of the circle, and in the range of the neighborhood with the $k$-value, the label with the most occurrences was used as the test sample label. Therefore, when the sample size is unbalanced, the classification results tend to be biased towards large-volume samples (Fig 5).

In Fig 5, the red hexagon is the test sample, and the training sample contains two kinds of labels: square and triangle. When $k = 5$, in the $k$ neighborhood, the classification result is biased toward large-capacity samples (triangles), resulting in the final labeling of the test sample as a triangle label (the test sample is closer to a square label).

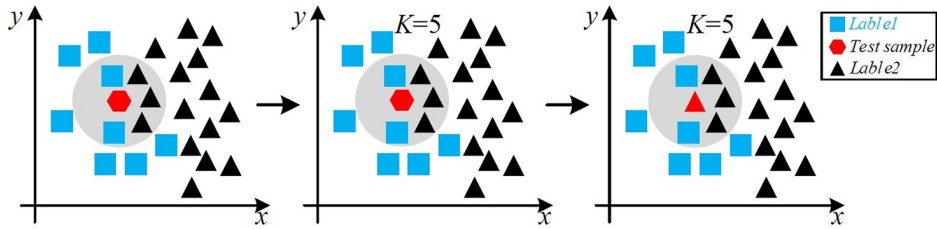

**Fig 5. Majority voting criteria.**

The original algorithm is improved based on the distance weight to balance the deviation of the classification results caused by the sample size. Assuming that there are $J$ classes in the datasets, Eq (21) is modified as:

$$Class = f\left(\max\left\{\left(1 - \frac{S_j}{S}\right)a_j\right\}\right), j = 1, \cdots, J \tag{26}$$

Where $Class$ is the predicted test sample class, $S$ is the total number of data, $S_j$ is the number of the $j$-th class of data, $a_i$ is the number of the selected KNN belonging to the $j$-th class, $f$ indicates that the class corresponding to the solution result can be returned. The test sample in Fig 4 is predicted using Eq (26) and result is a square.

$$
\begin{aligned}
Class \quad &= f\left(\max\left\{\left(1 - \frac{S_1}{S}\right)a_1, \left(1 - \frac{S_2}{S}\right)a_2\right\}\right) \\
&= f\left(\max\left\{\left(1 - \frac{8}{25}\right) \times 2, \left(1 - \frac{16}{25}\right) \times 3\right\}\right) \\
&= f\left(\max\left\{\frac{34}{25}, \frac{27}{25}\right\}\right) = f\left(\frac{34}{25}\right) = \blacksquare
\end{aligned} \tag{27}
$$

### 3.3. Time complexity analysis of IKNN

The traditional KNN algorithm does not require training and can be directly used for testing. So, its time complexity is $\mathcal{O}(k \times l \times m \times n)$, Among them, $k$ is the number of nearest neighbors, $n$ is the number of samples, $m$ is the sample feature dimension, and $l$ is the number of samples to be tested.

The improved KNN algorithm needs to be trained in the original data set to find the best metric matrix $\mathbf{M}$, and then tested. The training time complexity is $\mathcal{O}(k \times T \times m^2 \times n^2)$, and the testing time complexity is $\mathcal{O}(k \times l \times m^2 \times n)$, where $k$ is the number of nearest neighbors, $T$ is the number of training times, $n$ is the number of samples, $m$ is the sample feature dimension, and $l$ is the number of samples to be tested.

**Algorithm 3. The solution of weight matrix M.**

```
Input: T₀:initial temperature, Tₑ:Termination temperature
1: Tₖ = T₀
2: M₀ = I
3: while (Tₖ ≤ Tₑ)
4:   for1 i = 2:n
5:     for2 j = 1:i-1
6:         M(i,j+1) = |M(i,j)+RRᵀ / M(i,j)+RRᵀ| (R = rand(n,n) ∈ ℝⁿ)
```

```
 7:        ΔE = Z(M) - Z(M₀)
 8:        p₁ = min{1, e^(-ΔE/Tₖ)}.
 9:        if (p₁ > rand())
10:         M₀(i, j) = M(i, j +1)
11:         end if
12:      end for2
13:      end for1
14:    Tₖ₊₁ = ηTₖ (k ← k + 1)
15: end while
Return:M = M₀
```

## 4. Improved WOA and improved KNN for FS

### 4.1. Fitness function

Combining improved WOA and improved KNN algorithm for FS is a wrapper-based FS method. The main purpose of IWOA for FS is to find the smallest feature subset. IKNN is based on feature subsets to classify and get the best classification accuracy. Therefore, the FS method based on IWOA and IKNN has two goals: (1) Finding the least feature subset as much as possible; (2) Making the algorithm classification accuracy as high as possible in the found feature subset.

In order to solve these two conflicting goals, IWOA is regarded as a solution. The location of each individual whale is the solution. When the number of features in the solution is less, the classification accuracy is higher, and the solution is better [26]. In order to balance the two conflicting goals (the smallest number of features and the largest classification accuracy), the fitness function [64] is:

$$Fitness = \alpha\gamma_R(D) + \beta\frac{|R|}{|C|} \tag{28}$$

where $\gamma_R(D)$ represents the IKNN classification error rate, $|R|$ represents the length of the selected feature subset, $|C|$ represents the total number of features in the datasets, $\alpha \in (0, 1)$ represents the importance of classification quality, and $\beta = (1-\alpha)$ represents the importance of the length of the subset [64].

In order to make the proposed algorithm suitable for FS, this paper maps the continuous search space to the binary space. The main method is to take 1 when the algorithm fitness is greater than 0.5, and take 0 when it is less than or equal to 0.5.

$$x_{binary} = \begin{cases} 0, & fitness \leq 0.5 \\ 1, & fitness > 0.5 \end{cases} \tag{29}$$

### 4.2. Pseudo-code of the IWOAIKFS algorithm

In order to use IWOAIKFS for FS, a binary representation is used to represent the solution to the FS problem. Assuming that the selected feature is 1, and the unselected feature is 0. Therefore, the pseudo-code based on IWOAIKFS is shown in Algorithm 4.

### 4.3. Time complexity analysis of IWOAIKFS

Since IWOAIKFS is a feature selection method obtained by IWOA optimizing IKNN, its time complexity can be divided into three parts, namely IWOA time complexity, IKNN complexity and IWOA optimizing IKNN time complexity. So, the time complexity of IWOAIKFS can be obtained as $\mathcal{O}(k \times t_{max} \times (N \times m^2 \times n \times l + m^2 \times n^2) + t_{max})$, where $k$ is the number of

nearest neighbors, $N$ is the population size, $t_{\max}$ is the number of iterations, $n$ is the number of samples, $m$ is the sample feature dimension, and $l$ is the number of samples to be tested.

## 5. Experimental evaluation and discussion

In order to verify the effectiveness of the three methods proposed in this paper, three numerical experiments are designed as follows:

(1) In order to verify that IWOA has better optimization and convergence performance, this paper selects 8 benchmark functions and conducts simulation experiments in different dimensions (30 dimensional and 100 dimensional), and compare and analyze the optimization results of 8 meta-heuristic algorithms. The details are shown in Section 5.3.

```
Algorithm 4. IWOAIKFS.
Input: Dataset, t_max, lb, ub
1: Dataset normalization and Initialize a feature population X_i with
Algorithm 1
2: Calculate the fitness of each feature vector (Eq (28))
3: X_* = the best feature vector
4: while (t<t_max)
5:   for each individual
6:     Update a (Eq 15), A, C (Eq 16)
7:     Calculate p (Eqs 13 and 14)
8:     if1 (p<0.5)
9:       if2 (|A|<1)
10:          Update the position of current feature vector (Eq (17))
11:      else if2 (|A|≥1)
12:          Select a random search feature (X_rand)
13:          Update the position of current feature vector (Eq (1))
14:       end if2
15:     else if1 (p ≥ 0.5)
16:        Update the position of current feature vector (Eq (19))
17:     end if1
18:   end for
19:   Check the boundary and amend it
20:   Calculate the fitness of each feature vector
21:   Update X_* if there is a better feature vector
22:   t = t+1
23: end while
24: Divide X_* into training set and test set
25: Calculate the weight matrix M according to Algorithm 3
26: Calculate the distance between the test sample and the training
sample according
to Eq (23)
27: Sort each distance found
28: Select k points as the KNN of the test sample
29: Calculate and predict the test sample class (Eq (26))
30: Count the number of correct predictions and calculate Acc
Return: Accuracy (Acc)
```

(2) In order to verify that IKNN has better classification performance, this paper selects 8 datasets on UCI [65] for simulation experiments, and compare and analyze with the experimental results of 5 kinds of classifiers. The details are shown in Section 5.4.

(3) In order to verify that IWOAIKFS has good performance, this paper adds 7 datasets (15 datasets in total) to perform numerical experiments based on Section 5.4, and compares

**Table 1. Benchmark functions used and their details.**

| F | Index | Optimum | lb | ub |
|---|---|---|---|---|
| F1 | Sphere | 0 | -100 | 100 |
| F2 | Schwefel 2.22 | 0 | -10 | 10 |
| F3 | Schwefel 1.2 | 0 | -100 | 100 |
| F4 | Step | 0 | -100 | 100 |
| F5 | Rastrigin | 0 | -5.12 | 5.12 |
| F6 | Ackley | 0 | -32 | 32 |
| F7 | Weierstrass | 0 | -0.5 | 0.5 |
| F8 | Penalized | 0 | -50 | 50 |

and analyzes the experimental results with 6 FS methods based on the meta-heuristic algorithm. The details are shown in Section 5.5.

## 5.1. Experimental environment and datasets

### 5.1.1. Experimental environment. System: 64bit Windows 10

CPU: Intel(R) Core (TM) i7-5557U

Main frequency: 3.10GHz; RAM: 8G

Platform: Matlab2020b and Python 3.9

### 5.1.2. Benchmark functions and datasets. In order to evaluate the superior performance of IWOA, IKNN and IWOAIKFS, 8 benchmark functions and 15 datasets were selected for numerical experiments. The detailed description shown in Tables 1 and 2.

## 5.2. IWOA comparative experiment

### 5.2.1. Parameter setup for IWOA and other algorithms. In order to ensure the objective fairness of IWOA numerical experiments, the maximum number of iterations of all algorithms

**Table 2. Datasets used and their details.**

| Number | Datasets | Class | Features | Samples |
|---|---|---|---|---|
| 1 | Birds | 6 | 10 | 413 |
| 2 | Blood | 2 | 5 | 748 |
| 3 | Breast_cancer | 2 | 30 | 569 |
| 4 | Bupa | 2 | 6 | 345 |
| 5 | Car | 4 | 6 | 1728 |
| 6 | Chart | 6 | 60 | 600 |
| 7 | Digits | 10 | 64 | 1797 |
| 8 | Glass | 6 | 9 | 214 |
| 9 | Heart_disease | 2 | 13 | 303 |
| 10 | Indian | 2 | 10 | 583 |
| 11 | Ionosphere | 2 | 34 | 351 |
| 12 | Iris | 3 | 4 | 150 |
| 13 | Planning | 2 | 12 | 182 |
| 14 | Wine | 3 | 13 | 178 |
| 15 | Zoo | 7 | 16 | 101 |

**Table 3. Parameter settings of IWOA and other selected algorithms.**

| Algorithms | Parameters setting |
|---|---|
| IWOA | $b = 1$; $a$ decrease nonlinearly from 2 to 0 |
| WOA [15] | $b = 1$; $a$ decrease linearly from 2 to 0 |
| ASO [66] | $\alpha = 5$, $\beta = 0.2$ |
| GWO [18] | $a$ decrease linearly from 2 to 0 |
| HHO [67] | $E1$ decreases linearly from 2 to 0 |
| MFO [68] | $a$ decrease linearly from -1 to -2 |
| MVO [69] | $WEP$ increases linearly from 0.2 to 1 |
| | $TDR$ decreases nonlinearly from 0.7 to 0 |
| SSA [70] | $c_1$ decreases nonlinearly from 2 to 0 |
| TSA [71] | $x_{min} = 1$, $x_{max} = 4$; $A_1$ is a randomly generated number -1 to 1 |

was set to 1000. The initial population size of the algorithm was 50. Each group of experiments was performed 30 times, and the mean and standard deviation (STD) were calculated as the algorithm Evaluation indicators. The detailed parameter settings of the algorithm are shown in Table 3.

**5.2.2. IWOA experimental results and analysis.** Numerical experiments were carried out on 8 test functions in 30 and 100 dimensions, respectively. The mean and standard deviation of the optimization results of each algorithm on the test function was counted (Table 4). Fig 5 is the log mean convergence curve of F1, F3, F5, and F7 under different algorithms. Fig 6 shows a histogram of the average running time of different algorithms on 8 benchmark functions.

Table 4 shows that IWOA has a smaller mean and standard deviation on most test functions and shows better optimization results. The experimental results of IWOA were analyzed in 30 and 100 dimensions. Except for the F8 function inferior to the HHO algorithm, the IWOA's experimental results on the eight benchmark functions are several or even hundreds of orders of magnitude higher than the other comparison algorithms. The experimental results showed that IWOA had high convergence accuracy and stability. IWOA had high optimization results on unimodal functions and most multimodal functions, verifying that IWOA had better local mining and global exploration capabilities. Therefore, IWOA has better stability and convergence than the other 8 meta-heuristic algorithms.

In order to better compare the convergence performance of the 9 algorithms, the log-mean fitness values of the two unimodal functions (F1 and F3) and the two multimodal functions (F5 and F7) were selected to draw the convergence curve (Fig 6). Fig 6(a)–6(d) are F1, F3, F5, and F7 function images, respectively, and Fig 6(e)–6(h) and Fig 6(i)–6(l) correspond to F1, F3, F5, and F7, respectively Function, the average convergence curve of the number in 30 dimensions and 100 dimensions. Fig 6 shows that in the same dimension, the optimization of IWOA for unimodal functions is higher than that of the other 9 comparison algorithms. When the dimensions are different, IWOA has better convergence when optimizing high-dimensional multimodal functions. As the dimensionality increases, IWOA shows better convergence performance.

Fig 7(a) and 7(b) show the average running time of 9 algorithms in 30 and 100 dimensions, respectively. Fig 7 shows that the average running time of the IWOA algorithm is only better than that of the ASO algorithm due to the increased time of IWOA when initializing the population and generating skew random probabilities. However, the comparison of Fig 7(a) and 7(b) shows that the average running time increase of IWOA in high and low dimensions is smaller than that of other comparison algorithms with better convergence accuracy.

**Table 4. The comparison of obtained solutions for 8 benchmark functions.**

| F | Dim | Index | ASO | GWO | HHO | MFO | MVO | SSA | TSA | WOA | IWOA |
|---|---|---|---|---|---|---|---|---|---|---|---|
| F1 | 30 | Mean | 2.02E-23 | 3.62E-70 | 6.97E-193 | 2.33E+03 | 1.70E-01 | 8.55E-09 | 1.72E-51 | 4.20E-166 | **0.00E+00** |
| | | STD | 1.57E-23 | 1.04E-69 | 0.00E+00 | 5.04E+03 | 4.80E-02 | 1.83E-09 | 8.53E-51 | 0.00E+00 | **0.00E+00** |
| | 100 | Mean | 7.09E-03 | 2.28E-34 | 8.52E-191 | 4.12E+03 | 2.32E+01 | 6.18E-03 | 1.78E-27 | 1.97E-170 | **0.00E+00** |
| | | STD | 3.86E-02 | 3.68E-34 | 0.00E+00 | 1.68E+04 | 3.38E+00 | 5.12E-03 | 6.97E-27 | 0.00E+00 | **0.00E+00** |
| F2 | 30 | Mean | 4.89E-11 | 6.08E-41 | 1.61E-101 | 3.47E+01 | 2.89E-01 | 5.22E-01 | 1.29E-31 | 1.81E-107 | **4.90E-324** |
| | | STD | 4.71E-11 | 7.22E-41 | 7.34E-101 | 2.18E+01 | 7.59E-02 | 4.76E-01 | 1.16E-31 | 9.56E-107 | **0.00E+00** |
| | 100 | Mean | 3.07E+00 | 6.53E-21 | 4.45E-100 | 1.02E+02 | 3.48E+15 | 1.17E+01 | 1.07E-17 | 1.60E-105 | **1.33E-321** |
| | | STD | 1.67E+00 | 2.90E-21 | 2.14E-99 | 5.60E+01 | 1.75E+16 | 3.66E+00 | 1.42E-17 | 5.48E-105 | **0.00E+00** |
| F3 | 30 | Mean | 2.21E+02 | 5.90E-19 | 3.92E-156 | 1.79E+04 | 1.89E+01 | 4.32E+01 | 7.66E-16 | 9.93E+03 | **5.60E-320** |
| | | STD | 1.33E+02 | 2.17E-18 | 2.15E-155 | 1.21E+04 | 9.49E+00 | 4.22E+01 | 3.58E-15 | 5.85E+03 | **0.00E+00** |
| | 100 | Mean | 3.14E+04 | 6.59E-01 | 6.49E-142 | 9.71E+04 | 3.14E+04 | 2.24E+04 | 1.04E+03 | 7.19E+05 | **6.40E-264** |
| | | STD | 5.96E+03 | 2.17E+00 | 2.91E-141 | 6.41E+04 | 4.76E+03 | 9.84E+03 | 1.45E+03 | 1.12E+05 | **0.00E+00** |
| F4 | 30 | Mean | 0.00E+00 | 0.00E+00 | 0.00E+00 | 1.33E+03 | 5.70E+00 | 8.23E+00 | 3.78E+00 | 3.33E-02 | **0.00E+00** |
| | | STD | 0.00E+00 | 0.00E+00 | 0.00E+00 | 3.46E+03 | 2.81E+00 | 4.66E+00 | 7.23E-01 | 1.83E-01 | **0.00E+00** |
| | 100 | Mean | 3.48E+01 | 0.00E+00 | 0.00E+00 | 3.23E+03 | 1.43E+02 | 2.52E+02 | 1.37E+01 | 0.00E+00 | **0.00E+00** |
| | | STD | 4.46E+01 | 0.00E+00 | 0.00E+00 | 1.64E+04 | 4.77E+01 | 6.09E+01 | 1.19E+00 | 0.00E+00 | **0.00E+00** |
| F5 | 30 | Mean | 2.64E+01 | 6.87E-02 | 0.00E+00 | 1.42E+02 | 1.07E+02 | 4.54E+01 | 1.70E+02 | 0.00E+00 | **0.00E+00** |
| | | STD | 5.60E+00 | 3.76E-01 | 0.00E+00 | 4.54E+01 | 1.84E+01 | 1.54E+01 | 3.70E+01 | 0.00E+00 | **0.00E+00** |
| | 100 | Mean | 1.29E+02 | 1.17E-13 | 0.00E+00 | 5.59E+02 | 6.16E+02 | 1.32E+02 | 9.30E+02 | 0.00E+00 | **0.00E+00** |
| | | STD | 1.84E+01 | 8.70E-14 | 0.00E+00 | 7.81E+01 | 8.31E+01 | 3.87E+01 | 1.18E+02 | 0.00E+00 | **0.00E+00** |
| F6 | 30 | Mean | 3.28E-12 | 1.28E-14 | 8.88E-16 | 1.11E+01 | 8.99E-01 | 1.92E+00 | 2.04E+00 | 3.85E-15 | **8.88E-16** |
| | | STD | 1.83E-12 | 2.72E-15 | 0.00E+00 | 9.07E+00 | 8.71E-01 | 8.15E-01 | 1.48E+00 | 2.30E-15 | **0.00E+00** |
| | 100 | Mean | 1.39E+00 | 6.95E-14 | 8.88E-16 | 1.86E+01 | 4.47E+00 | 5.52E+00 | 4.41E-14 | 3.26E-15 | **8.88E-16** |
| | | STD | 5.55E-01 | 6.13E-15 | 0.00E+00 | 3.84E-01 | 4.17E+00 | 1.14E+00 | 1.24E-14 | 2.53E-15 | **0.00E+00** |
| F7 | 30 | Mean | 3.26E-03 | 0.00E+00 | 0.00E+00 | 4.96E+00 | 9.31E+00 | 1.11E+01 | 2.13E-15 | 0.00E+00 | **0.00E+00** |
| | | STD | 1.30E-02 | 0.00E+00 | 0.00E+00 | 2.93E+00 | 2.86E+00 | 2.51E+00 | 4.63E-15 | 0.00E+00 | **0.00E+00** |
| | 100 | Mean | 1.32E+01 | 3.22E-14 | 0.00E+00 | 5.99E+01 | 8.10E+01 | 6.18E+01 | 1.94E-09 | 0.00E+00 | **0.00E+00** |
| | | STD | 3.77E+00 | 2.21E-14 | 0.00E+00 | 7.55E+00 | 6.98E+00 | 5.21E+00 | 1.07E-08 | 0.00E+00 | **0.00E+00** |
| F8 | 30 | Mean | 8.86E-03 | 2.73E-02 | **1.17E-06** | 2.22E-01 | 1.20E+00 | 4.20E+00 | 7.93E+00 | 1.67E-03 | 4.69E-03 |
| | | STD | 2.79E-02 | 2.16E-02 | **1.67E-06** | 3.49E-01 | 1.11E+00 | 2.44E+00 | 3.68E+00 | 2.10E-03 | 1.75E-02 |
| | 100 | Mean | 1.99E+00 | 1.86E-01 | **1.81E-07** | 4.61E+05 | 9.90E+00 | 1.13E+01 | 1.00E+01 | 5.73E-03 | 6.26E-03 |
| | | STD | 7.17E-01 | 5.15E-02 | **2.51E-07** | 1.45E+08 | 2.27E+00 | 2.82E+00 | 2.74E+00 | 2.50E-03 | 2.28E-03 |

Compared with the other 8 comparison algorithms, IWOA has better local convergence and global optimization and shows better reliability and robustness than other comparison algorithms when solving high-dimensional multimodal functions.

From Table 4, Figs 6 and 7 shows that when IWOA optimizes the 8 benchmark functions, the smaller mean and standard deviation feedback that IWOA has better overall optimization stability. The convergence curve shows that IWOA has better optimization results on high-dimensional multimodal functions. The average running time of IWOA is higher than that of the comparison algorithm. However, IWOA can achieve higher accuracy with the sacrifice of a small increase in time. Therefore, IWOA has better optimization performance than the eight comparison algorithms.

**5.2.3. Wilcoxon's test and Friedman test.** Only the mean and standard deviation of the results of 30 independent experiments cannot fully measure the superiority of the improved algorithm. As one of the nonparametric statistical test methods for evaluating the performance

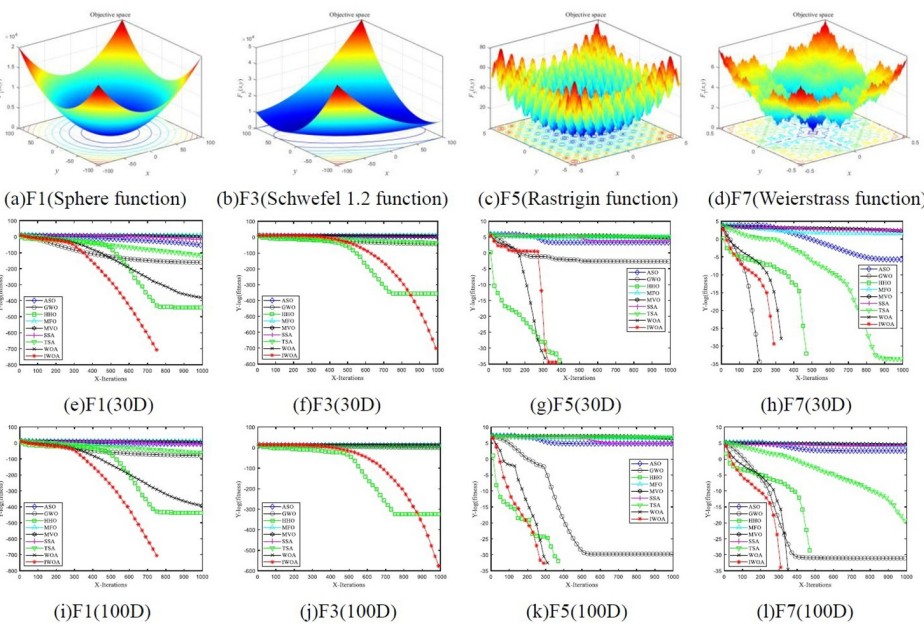

**Fig 6. Logarithmic mean convergence curves of different algorithms.** (a)F1(Sphere function), b)F3(Schwefel 1.2 function), (c)F5(Rastrigin function), d)F7(Weierstrass function), (e)F1(30D), (f)F3(30D), (g)F5(30D), (h)F7(30D). (i) F1(100D), (j)F3(100D), (k)F5(100D), (l)F7(100D).

of algorithms, Wilcoxon's rank sum test [72] is often used to verify the performance of meta-heuristic algorithms Wilcoxon's test was used to conduct experiments at the 5% significance level to judge whether each result of IWOA was statistically significantly different from the best results of other algorithms. Table 5 shows the *p*-values calculated in the IWOA of the eight benchmark functions and the Wilcoxon's test of the other algorithms. $p < 0.05$ is considered as a strong verification to reject the null hypothesis.

Table 5 shows that the *p*-value of IWOA is less than 0.05 at 30 dimensions, indicating that the optimization performance of IWOA is statistically significant and verifying that IWOA has higher convergence accuracy than other comparison algorithms. The *p*-value of IWOA is second only to the comparison with the F4 and F5 functions of WOA in 100 dimensions, showing IWOA has a better performance compared with other algorithms.

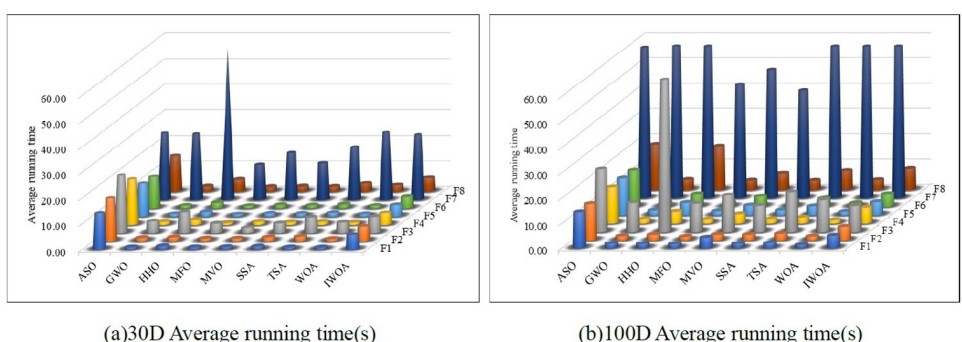

**Fig 7. Average running time of different algorithms.** (a) 30D Average running time(s), (b) 100D Average running time(s).

**Table 5. *p*-value of the Wilcoxon test for the optimization results of IWOA and other algorithms based 8 benchmark functions (*p*> = 0.5 are in bold).**

| F | IWOA vs. ASO | IWOA vs. GWO | IWOA vs. HHO | IWOA vs. MFO | IWOA vs. MVO | IWOA vs. SSA | IWOA vs. TSA | IWOA vs. WOA |
|---|---|---|---|---|---|---|---|---|
| | | | | D = 30 | | | | |
| F1 | 2.06E-184 | 5.21E-41 | 1.31E-25 | 3.94E-317 | 1.81E-310 | 2.60E-281 | 1.04E-84 | 1.15E-33 |
| F2 | 4.85E-233 | 3.66E-40 | 2.64E-35 | 9.72E-320 | 8.75E-315 | 4.15E-308 | 3.64E-86 | 1.75E-30 |
| F3 | 2.78E-64 | 8.08E-05 | 9.71E-19 | 1.86E-119 | 1.57E-72 | 7.67E-62 | 1.19E-10 | 3.47E-171 |
| F4 | 8.62E-05 | 4.04E-24 | 2.07E-48 | **NA** | **NA** | **NA** | 9.82E-305 | 2.50E-149 |
| F5 | 3.71E-305 | 1.64E-89 | 4.16E-03 | 1.99E-321 | **NA** | 1.23E-312 | 1.00E-323 | 2.07E-01 |
| F6 | 3.70E-199 | 5.01E-68 | 1.65E-13 | 1.15E-320 | 5.31E-314 | 7.93E-310 | 1.27E-303 | 9.71E-84 |
| F7 | 4.80E-293 | 1.98E-04 | 1.63E-17 | **NA** | **NA** | **NA** | 1.59E-185 | 7.24E-04 |
| F8 | 5.59E-153 | 5.08E-173 | 5.85E-312 | 9.30E-287 | 2.37E-290 | 1.11E-291 | 4.03E-287 | 9.36E-119 |
| w\|t\|l | 8\|0\|0 | 8\|0\|0 | 8\|0\|0 | 6\|2\|0 | 5\|3\|0 | 6\|2\|0 | 8\|0\|0 | 8\|0\|0 |
| | | | | D = 100 | | | | |
| F1 | 4.06E-274 | 3.15E-79 | 3.66E-22 | 1.20E-322 | 2.25E-315 | 2.65E-297 | 1.31E-129 | 6.81E-30 |
| F2 | 1.35E-306 | 1.74E-80 | 1.06E-34 | 6.29E-321 | 5.90E-323 | 1.28E-314 | 9.62E-134 | 7.85E-31 |
| F3 | 3.43E-58 | 1.53E-28 | 4.24E-27 | 3.57E-80 | 3.74E-67 | 1.18E-57 | 8.70E-43 | 4.88E-223 |
| F4 | **NA** | 1.36E-02 | 1.57E-43 | **NA** | **NA** | **NA** | 1.18E-310 | **0.16992** |
| F5 | 5.63E-322 | 5.30E-148 | 9.53E-02 | **NA** | **NA** | **NA** | **NA** | **0.07075** |
| F6 | 4.18E-304 | 9.30E-87 | 6.37E-15 | **NA** | 6.30E-320 | 1.64E-316 | 4.27E-143 | 2.64E-83 |
| F7 | **NA** | 4.01E-95 | 6.86E-17 | **NA** | **NA** | **NA** | 9.48E-226 | 4.75E-02 |
| F8 | 8.75E-283 | 2.56E-268 | 1.22E-313 | 1.78E-321 | 9.90E-307 | 9.53E-296 | 1.44E-289 | 2.96E-31 |
| w\|t\|l | 6\|2\|0 | 8\|0\|0 | 8\|0\|0 | 4\|4\|0 | 5\|3\|0 | 5\|3\|0 | 7\|1\|0 | 6\|0\|2 |

In order to better evaluate the effectiveness of the method proposed in this paper and better detect the significant differences between two or more observation data, the method Friedman is used to perform statistical tests on the algorithm proposed in this paper. Friedman test is a nonparametric two-way analysis of variance method [72], The test process is as follows:

(1) Collect observations for each algorithm or problem;

(2) Each question is ranked from the best result (1) to the worst result (*k*), the *i*-th question is defined as; $r_i^j (1 \leq j \leq k)$;

(3) Find the average ranking of each algorithm in all problems, and get the final ranking

$$R_j = \frac{1}{n} \sum_{i=1}^{n} r_i^j.$$

Under the null hypothesis, the rank $R_j$ of all algorithms is equal, and the Friedman statistic value $F_f$ is shown in formula (30), According to Table 4, the Friedman test was implemented on Matlab 2020b for IWOA and the comparison algorithm, and the results are shown in Table 6.

$$F_f = \frac{12n}{k(k+1)} \left[ \sum_j R_j^2 - k(k+1)^2/4 \right] \tag{30}$$

It can be seen from Table 6 that the asymptotically significant *p*-value obtained by the Friedman test is far less than 0.01 on both the 30D and 100D benchmark functions, so it can be seen that IWOA is comparable to the 30D and 100D benchmark functions. There are significant differences between the algorithms, Therefore, it can be seen that there are significant differences between IWOA and the comparison algorithms in both 30D and 100D benchmark

**Table 6. Friedman test results of benchmark functions with different dimensions.**

| Dim | p-value | IWOA | ASO | GWO | HHO | MFO | MVO | SSA | TSA | WOA |
|------|----------|------|------|------|------|------|------|------|------|------|
| 30D | 6.28E-08 | 1.38 | 5.38 | 3.88 | 1.88 | 8.25 | 6.88 | 7.38 | 6.38 | 3.63 |
| 100D | 3.03E-08 | 1.25 | 5.88 | 4.00 | 2.00 | 8.25 | 7.63 | 7.00 | 5.50 | 3.50 |

functions; However, the rank mean of the IWOA algorithm is the smallest in both 30D (1.38) and 100D (1.25), indicating that its optimization performance is the best. Combining the test results of Wilcoxon's test and Friedman test, it can be concluded that IWOA has better performance than the comparison algorithm in general.

## 5.3. IKNN comparative experiment

**5.3.1. Influence of 3 strategies on KNN algorithm.** The KNN algorithm that is only affected by the weight matrix **M** is MKNN. The KNN algorithm that is only affected by the weighted classification criteria is WKNN. The KNN algorithm affected by the two strategies is IKNN.

The simulated annealing temperature $T_0 = 100$, the end temperature $T_e = 1$, the number of cycles per temperature is 100, and the learning rate $\eta = 0.9$, then the number of iterations required to generate M is:

$$t = 100 \times \frac{\ln T_e - \ln T_0}{\ln \eta} \approx 4370.869 \approx 4400 \tag{31}$$

8 datasets in Section 5.1 were taken, and the numerical experiments were performed in Python 3.9 environment. The experimental results under different strategies are shown in Table 5 (Breast: Breast_cancer, Heart: Heart_disease).

**5.3.2. Comparison of classification accuracy between IKNN and other classifiers.** In order to further verify the effectiveness of IKNN, we uses KNN [62], Naive Bayes [73], C4.5 [74], SVM [75] and BP neural network [76] as comparison algorithms for numerical experiments. Among them, the number of nearest neighbors of KNN and IKNN $k = 10$. SVM uses Gaussian kernel function. BP neural network uses a stochastic gradient optimizer, sets 1 hidden layer. The number of hidden layer nodes is 13, and the activation function is a linear function. The experimental results of each algorithm on the datasets in the Python 3.9 environment are shown in Table 8.

**5.3.3. IKNN results discussion.** Table 7 shows the experimental results of the KNN algorithm on 8 datasets under different strategies. Under the same parameter settings, the

**Table 7. The comparison of classification accuracy of different strategies.**

| Datasets | Accuracy | | | |
|----------|------|------|------|------|
| | KNN | MKNN | WKNN | IKNN |
| Birds | 0.6946 | 0.7665 | 0.7186 | **0.7844** |
| Breast | 0.9735 | 0.9779 | 0.9779 | **0.9779** |
| Digits | 0.9817 | 0.9848 | 0.9772 | **0.9863** |
| Glass | 0.7089 | 0.7722 | 0.7215 | **0.7975** |
| Heart | 0.8148 | 0.8426 | 0.8148 | **0.8426** |
| Iris | 0.9286 | 0.9464 | 0.9286 | **0.9464** |
| Ionosphere | 0.8571 | 0.8947 | 0.8722 | **0.9173** |
| Wine | 0.9452 | 0.9589 | 0.9452 | **0.9589** |

**Table 8. The comparison of classification accuracy of different classifiers.**

| Datasets | Accuracy | | | | | |
|---|---|---|---|---|---|---|
| | KNN | Naive bayes | C4.5 | SVM | BP neural network | IKNN |
| Birds | 0.6946 | 0.4491 | 0.5868 | **0.8383** | 0.6647 | 0.7844 |
| Breast | 0.9735 | 0.9690 | 0.9336 | 0.6858 | 0.9779 | **0.9779** |
| Digits | 0.9817 | 0.7610 | 0.8234 | 0.8219 | 0.9619 | **0.9863** |
| Glass | 0.7089 | 0.4304 | 0.6076 | 0.5190 | 0.6709 | **0.7975** |
| Heart | 0.8148 | 0.8056 | 0.7407 | 0.5093 | 0.8056 | **0.8426** |
| Iris | 0.9286 | 0.9107 | 0.9286 | 0.7857 | 0.9107 | **0.9464** |
| Ionosphere | 0.8571 | 0.8722 | 0.9173 | 0.6391 | 0.8872 | **0.9173** |
| Wine | 0.9452 | **1.0000** | 0.9178 | 0.3836 | 0.9589 | 0.9589 |

classification accuracy of MKNN, WKNN and IKNN on the 8 datasets is greater than or equal to that of the KNN algorithm. Among them, the IKNN algorithm that combines two strategies has the best effect. In the KNN algorithm improved based on a single strategy, MKNN performs better. Therefore, using the weight matrix **M** to measure the importance of samples has better classification performance than the simple Euclidean distance to measure the sample distance. When the amount of data is small, the classification accuracy of WKNN has no obvious change compared with the KNN algorithm, but for data sets with large amounts of data, WKNN has a better classification effect. The three proposed strategies can effectively improve the KNN algorithm and its classification performance.

Table 8 shows the classification experiment results of the IKNN algorithm and other 5 classifiers on 8 datasets. Table 8 show that in the same experimental environment, the classification accuracy of the IKNN algorithm on most datasets is higher than other comparison algorithms. Among them, the classification accuracy of the IKNN algorithm has been improved most significantly on the Glass dataset, and the classification accuracy on the Wine dataset and Bird dataset is inferior to Naive Bayes and SVM due to the universality of the algorithm. Naive Bayes and SVM are more suitable for Wine and Bord datasets (the effect is not obvious on other datasets). Therefore, IKNN has better classification performance than comparison algorithms.

## 5.4. IWOAIKFS comparative experiment

The programming tool MATLAB 2020b was applied to verify that IWOAIKFS has better classification performance. In the computing environment of Section 5.1, experiments were carried out using 15 datasets (Table 2) in UCI. The focus of the experiment is to use the IKNN classifier for IWOA ($k$ = 5 [8, 16, 20, 23, 26, 41, 54, 64, 77]) and the latest 6 meta-heuristic algorithms to use the KNN classifier ($k$ = 5), performance comparison of 30 independent experiments under 15 datasets.

The evaluation indicators of the experimental results are the mean classification accuracy of the algorithm for 30 independent experiments. The standard deviation of the optimal accuracy of the algorithm for 30 independent experiments, and the average number of features selected by the algorithm in 30 independent experiments.

**5.4.1. Parameter setup for IWOAIKFS and other optimizers.** Table 9:

**5.4.2. IWOAIKFS comparison of classification accuracy with other optimizers.** In order to test the effectiveness of IWOAIKFS, IWOAIKFS were compared with 6 FS methods according to meta-heuristic algorithms (ASO, GWO, HHO, SCA, SSA, and WOA). Table 10 shows the average accuracy, standard deviation, and the average number of selected features of

**Table 9. Parameter settings of IWOAIKFS and other selected algorithms.**

| Algorithms | Parameters setting |
|---|---|
| IWOAIKFS | Population Number (10); Maximum number of iterations (100); $b = 1$, $k = 5$ |
| | $a$ decrease nonlinearly from 2 to 0; Dimension corresponds to the number of features |
| WOA [78] | Population Number (10); Maximum number of iterations (100); $b = 1$, $k = 5$ |
| | $a$ decrease linearly from 2 to 0 |
| | Dimension corresponds to the number of features |
| ASO [22] | Population Number (10); Maximum number of iterations (100) |
| | $\alpha = 5$, $\beta = 0.2$, $k = 5$; Dimension corresponds to the number of features |
| GWO [79] | Population Number (10); Maximum number of iterations (100); $k = 5$ |
| | $a$ decrease linearly from 2 to 0 |
| | Dimension corresponds to the number of features |
| HHO [23] | Population Number (10); Maximum number of iterations (100); $k = 5$ |
| | $E1$ decreases linearly from 2 to 0 |
| | Dimension corresponds to the number of features |
| SCA [80] | Population Number (10); Maximum number of iterations (100); $\alpha = 2$, $k = 5$; Dimension corresponds to the number of features |
| SSA [81] | Population Number (10); Maximum number of iterations (100); $k = 5$ |
| | $c_1$ decreases nonlinearly from 2 to 0 |
| | Dimension corresponds to the number of features |

each algorithm for 30 independent experiments under 15 datasets. Under 30 independent experiments on 15 datasets, the comparison of IWOAIKFS between other optimizers is shown in Table 10:

1. In terms of average classification accuracy index, IWOAIKFS has the highest classification accuracy on 14 datasets, ranking first among 7 algorithms. The mean classification accuracy of IWOAIKFS on the Breast_cancer, Chart, Iris, Wine, and Zoo datasets has reached 100%. IWOAIKFS only ranks 2nd on the Heart_disease dataset, slightly inferior to the HHO algorithm. However, the standard deviation of IWOAIKFS is 0, indicating stronger stability relative to HHO. Therefore, IWOAIKFS has better classification accuracy on 15 datasets than other algorithms in terms of mean classification accuracy.

2. In the standard deviation indicator, the standard deviation of SCA on 15 datasets is all 0, ranking first among 7 algorithms, indicating that the algorithm is more stable. IWOAIKFS ranks 3rd among 7 algorithms, inferior to SCA and HHO algorithms, caused by the IKNN calculation of the weight matrix **M** for different datasets and the randomness of the algorithm.

3. In terms of selecting the number of features, IWOAIKFS chose the least features on the Breast_cancer, Car, Glass, Heart_disease and Ionosphere datasets, ranking 2nd among 7 algorithms, second only to the ASO algorithm (the least number of features is selected on 6 datasets).

Fig 8 shows the total average accuracy of IWOAIKFS on all data sets. Fig 9 shows a visualized bar graph of the average accuracy of IWOAIKFS and other algorithms in 15 datasets. Figs 8 and 9 identifies that in 15 datasets, IWOAIKFS has better performance in classification accuracy than other algorithms. The IWOAIKFS algorithm has the highest total mean accuracy in

**Table 10. Comparison between IWOAIKFS with other competitor optimizers based on accuracy ($k = 5$ and best are in bold).**

| Datasets | Index | ASO | GWO | HHO | SCA | SSA | WOA | IWOAIKFS |
|---|---|---|---|---|---|---|---|---|
| Birds | Accuracy | 0.8642 | 0.8622 | 0.8659 | 0.8630 | 0.8565 | 0.8480 | **0.8823** |
| | STD | 0.0042 | 0.0097 | **0.0000** | **0.0000** | 0.0105 | 0.0139 | 0.0139 |
| | Feature | **4.0333** | 4.1667 | 4.2667 | 4.0667 | 4.8000 | 5.0333 | 4.5333 |
| Blood | Accuracy | 0.8054 | 0.8054 | 0.8054 | 0.8054 | 0.8054 | 0.8040 | **0.8591** |
| | STD | **0.0000** | **0.0000** | **0.0000** | **0.0000** | **0.0000** | 0.0027 | 0.0041 |
| | Feature | **2.0000** | 2.5333 | 2.5000 | 2.4667 | 2.4667 | 2.4667 | 2.5333 |
| Breast_cancer | Accuracy | 0.9997 | 0.9988 | **1.0000** | 0.9976 | 0.9947 | 0.9935 | **1.0000** |
| | STD | 0.0016 | 0.0031 | **0.0000** | **0.0000** | 0.0044 | 0.0040 | **0.0000** |
| | Feature | 14.2333 | 16.5000 | 17.3667 | 14.2000 | 16.4000 | 17.2667 | **14.1000** |
| Bupa | Accuracy | 0.7536 | 0.7536 | 0.7536 | 0.7536 | 0.7536 | 0.7444 | **0.7761** |
| | STD | **0.0000** | **0.0000** | **0.0000** | **0.0000** | **0.0000** | 0.0291 | **0.0000** |
| | Feature | **2.0000** | 3.1667 | 2.9000 | 3.0667 | 2.8333 | 2.9000 | 2.7667 |
| Car | Accuracy | 0.9240 | 0.9246 | 0.9246 | 0.9098 | 0.9162 | 0.9202 | **0.9807** |
| | STD | 0.0037 | **0.0000** | **0.0000** | **0.0000** | 0.0198 | 0.0243 | 0.0070 |
| | Feature | 5.9667 | 6.0000 | 6.0000 | 5.6667 | 5.7000 | 5.9667 | **5.5333** |
| Chart | Accuracy | 0.9961 | 0.9939 | 0.9933 | 0.9925 | 0.9886 | 0.9858 | **1.0000** |
| | STD | 0.0042 | 0.0043 | 0.0034 | **0.0000** | 0.0064 | 0.0050 | **0.0000** |
| | Feature | 27.4000 | 27.5000 | 36.9333 | **21.7000** | 31.4667 | 38.1667 | 33.7333 |
| Digits | Accuracy | 0.9864 | 0.9851 | 0.9868 | 0.9735 | 0.9786 | 0.9843 | **0.9909** |
| | STD | 0.0034 | 0.0040 | 0.0018 | **0.0000** | 0.0036 | 0.0019 | 0.0030 |
| | Feature | 36.2000 | **30.8667** | 46.5667 | 31.0667 | 36.0667 | 58.4667 | 37.4667 |
| Glass | Accuracy | 0.8008 | 0.8024 | 0.8095 | 0.7960 | 0.7944 | 0.7802 | **0.8223** |
| | STD | 0.0117 | 0.0111 | **0.0000** | **0.0000** | 0.0159 | 0.0223 | 0.0258 |
| | Feature | 4.6000 | 5.0333 | 4.6000 | 4.8333 | 5.2667 | 5.7000 | **4.5000** |
| Heart_disease | Accuracy | 0.8839 | 0.8833 | **0.8911** | 0.8783 | 0.8533 | 0.8317 | 0.8865 |
| | STD | 0.0198 | 0.0284 | 0.0199 | **0.0000** | 0.0183 | 0.0323 | **0.0000** |
| | Feature | 5.3000 | 5.0000 | 5.6333 | 4.6667 | 6.6667 | 5.2333 | **4.6333** |
| Indian | Accuracy | 0.7773 | 0.7727 | 0.7807 | 0.7779 | 0.7644 | 0.7491 | **0.8098** |
| | STD | 0.0056 | 0.0107 | 0.0049 | **0.0000** | 0.0136 | 0.0210 | 0.0316 |
| | Feature | 3.0000 | 3.1667 | 3.1667 | **2.9000** | 3.8667 | 3.2333 | 4.0333 |
| Ionosphere | Accuracy | 0.9214 | 0.9286 | 0.9157 | 0.9452 | 0.9090 | 0.9210 | **0.9496** |
| | STD | 0.0082 | 0.0172 | 0.0115 | **0.0000** | 0.0109 | 0.0187 | 0.0111 |
| | Feature | 10.5000 | 11.3000 | 14.1333 | 5.5333 | 15.0000 | 7.0667 | **5.3000** |
| Iris | Accuracy | 0.9667 | 0.9667 | 0.9667 | 0.9667 | 0.9667 | 0.9667 | **1.0000** |
| | STD | **0.0000** | **0.0000** | **0.0000** | **0.0000** | **0.0000** | **0.0000** | **0.0000** |
| | Feature | **1.0000** | 2.6333 | 2.4333 | 2.6333 | 2.3000 | 2.3667 | 2.4333 |
| Planning | Accuracy | 0.8093 | 0.8046 | 0.8241 | 0.7991 | 0.7907 | 0.7769 | **0.8366** |
| | STD | 0.0189 | 0.0212 | 0.0133 | **0.0000** | 0.0250 | 0.0171 | 0.0243 |
| | Feature | 5.8000 | 6.1333 | 6.5667 | 5.0000 | 5.9667 | **4.5667** | 5.6667 |
| Wine | Accuracy | **1.0000** | **1.0000** | **1.0000** | **1.0000** | **1.0000** | 0.9971 | **1.0000** |
| | STD | **0.0000** | **0.0000** | **0.0000** | **0.0000** | **0.0000** | 0.0087 | **0.0000** |
| | Feature | **3.8667** | 6.7000 | 7.6667 | 6.5333 | 6.6000 | 8.2333 | 5.9333 |
| Zoo | Accuracy | 0.9983 | 0.9917 | **1.0000** | 0.9867 | 0.9850 | 0.9517 | **1.0000** |
| | STD | 0.0091 | 0.0190 | **0.0000** | **0.0000** | 0.0233 | 0.0404 | **0.0000** |
| | Feature | **7.3000** | 9.2333 | 9.8000 | 8.0000 | 9.4333 | 10.4000 | 9.9333 |

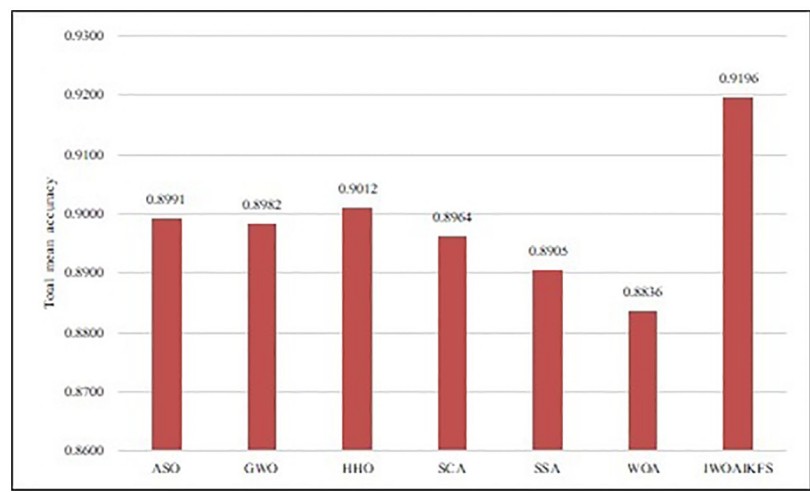

**Fig 8. Total mean accuracy of IWOAIKFS compared to other algorithms under all datasets.**

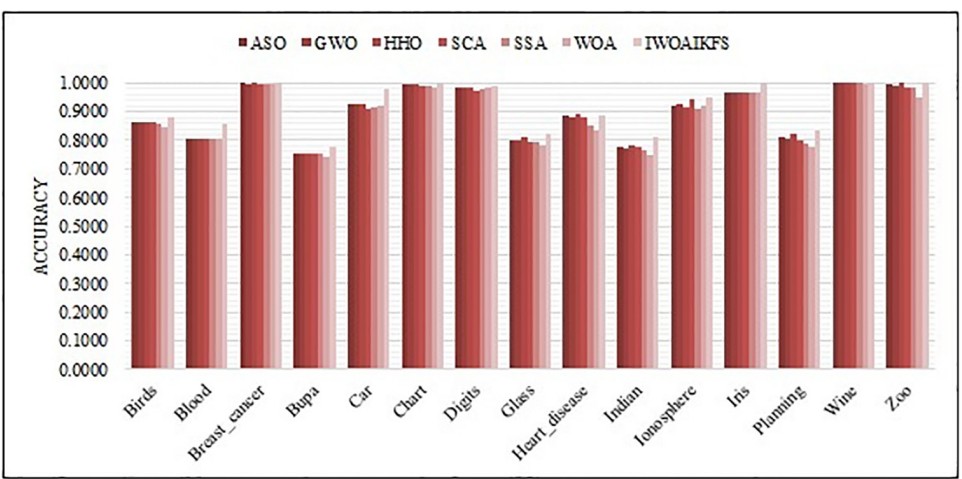

**Fig 9. The accuracy of IWOAIKFS compared to other optimizers.**

15 datasets, reaching 91.96%, 2.05%, 2.14%, 1.84%, 2.32%, 2.91%, 2.60% higher than 6 algorithms, respectively.

Fig 10 shows that each algorithm performs 30 independent experiments on 15 datasets and the ratio of the average number of features selected by each algorithm. IWOAIKFS is inferior to one or more of the algorithms on most data sets. However, from an overall point of view, most of the feature ratios selected by IWOAIKFS are below the average. Therefore, it can be considered to have better FS performance.

In summary, considering the three indicators (accuracy, standard deviation and number of FSs), IWOAIKFS has better classification accuracy on 15 datasets than other optimizers. It also exerts better effects on the standard deviation index and the average number of selected features than most algorithms. Therefore, IWOAIKFS can be considered to have better superior performance.

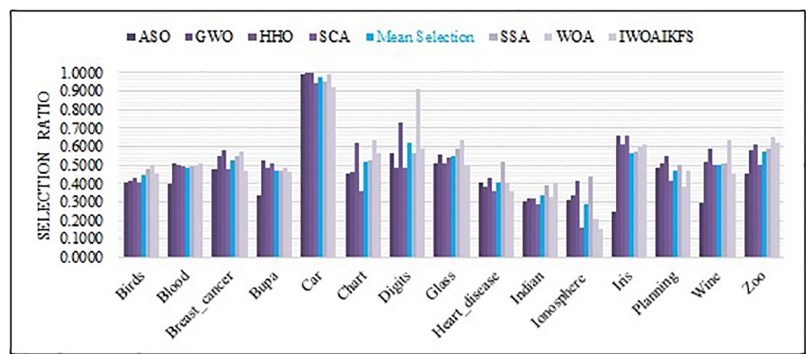

**Fig 10. The selection ratio of IWOAIKFS compared to other optimizers.**

**5.4.3. IWOAIKFS convergence comparison with other optimizers.** Fig 11 shows the mean fitness convergence curve of IWOAIKFS and the other 6 optimizers under 15 datasets. These algorithms are all tested and evaluated under the same population size and number of iterations.

Fig 11(a)–11(o) represents the mean fitness convergence curve of different datasets under 7 algorithms. Except for the poor convergence performance on the Breast_cancer and Wine datasets (Fig 11(c)–11(n)), IWOAIKFS shows better convergence performance in the remaining datasets. The convergence speed of the algorithm determines the final classification accuracy. IWOAIKFS shows the best fitness value on the Heart_disease dataset, but its classification accuracy does not perform well on the Heart_disease dataset. This is because under 100 iterations of IWOAIKFS, the Breast_cancer and Wine datasets have reached the optimal value, so their mean accuracy values have reached the optimal value. For the Heart_disease data set, when the convergence of IWOAIKFS reaches the best and IKNN evaluates the best subset, The constructed weight matrix **M** has random effects. Therefore, from the perspective of the convergence curve, IWOAIKFS shows better convergence performance overall.

**5.4.4. IWOAIKFS Wilcoxon's test and Friedman test.** Table 11 is the $p$-value of Wilcoxon's test based on the mean classification accuracy of 30 independent experiments. Compared with other optimizers, IWOAIKFS has better statistical significance on all datasets because its test $p$-value is less than 0.05. In addition, since its standard deviation is 0, ASO, GWO, HHO, SCA and SSA show the same statistical performance on the Blood, Bupa and Wine datasets; ASO, GWO, HHO, SCA, SSA and WOA show the same performance for Iris.

Table 12 is the $p$-value of Friedman test results. It can be seen from Table 12 that the asymptotically significant $p$-value obtained by the Friedman test is far less than 0.01 (3.66E-12), so it can be seen that there are significant differences between IWOAIKFS and the comparison algorithms on the 15 UCI benchmark data sets; but the rank of IWOAIKFS The mean is the smallest (1.53) among all contrasting algorithms, indicating that it has better optimization performance than contrasting algorithms.

Fig 12 shows a boxplot of classification accuracy obtained by all optimizers performing 30 independent experiments on 15 datasets. In Fig 12, the lower quartile ($Q_j$) represents lower values, the upper quartile ($Q_3$) represents higher values, and the red line in the box represents the median value. It can be seen from Fig 12 that IWOAIKFS ranks first in performance among all algorithms, and has the best performance in 15 datasets.

**5.4.5. IWOAIKFS results discussion.** Through the experimental results and analysis in Section 5.5, when the meta-heuristic algorithm is applied to the FS, IWOAIKFS has better

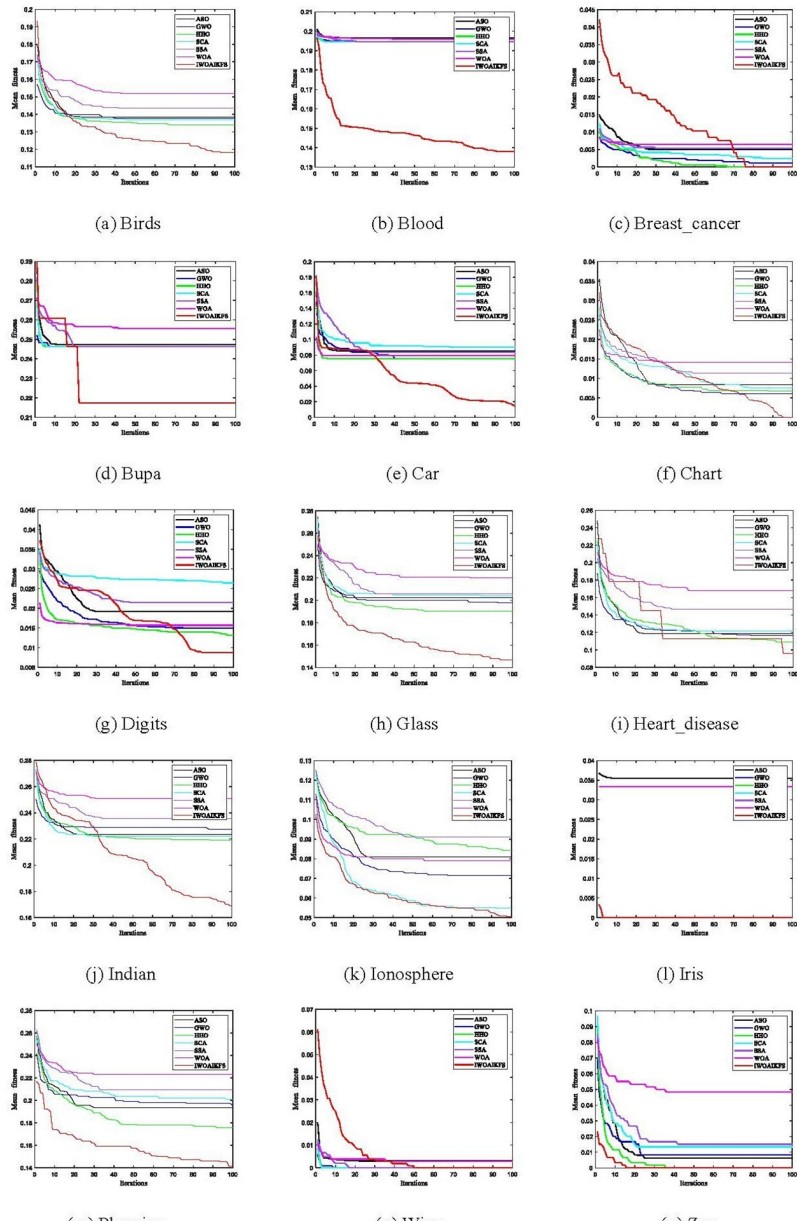

**Fig 11. Convergence curve of IWOAIKFS versus other algorithms over all datasets (k = 5).** (a) Birds, (b) Blood, (c) Breast_cancer, (d) Bupa, (e) Car, (f) Chart, (g) Digits, (h) Glass, (i) Heart_disease, (j) Indian, (k) Ionosphere, (l) Iris, (m) Planning, (n) Wine, (o) Zoo.

overall optimization performance than other meta-heuristic algorithms. First of all, considering only the mean classification accuracy of 30 experiments, IWOAIKFS has the best search performance, achieving a total mean classification accuracy of 91.96% for all datasets, at least 1.8% higher than other meta-heuristic algorithms, indicating that the proposed algorithm has higher classification performance. Secondly, considering only the accuracy standard deviation of 30 experiments, the standard deviation of SCA on all datasets is 0, indicating that SCA has better stability than other algorithms. Only from the analysis of the number of FS, ASO

**Table 11.** *p*-value of the Wilcoxon test for the classification accuracy results of IWOAIKFS and other optimizers ($k = 5$ and $p > = 0.5$ are in bold).

| IWOAIKFS vs. | ASO | GWO | HHO | SCA | SSA | WOA | ASO |
|---|---|---|---|---|---|---|---|
| Bird | 3.64E-12 | 5.72E-12 | 1.07E-12 | 5.72E-12 | 1.82E-11 | 2.17E-11 | 3.64E-12 |
| Blood | 1.13E-12 | 1.13E-12 | 1.13E-12 | 1.13E-12 | 1.13E-12 | 5.96E-12 | 1.13E-12 |
| Breast_cancer | 2.71E-14 | 8.64E-14 | 1.69E-14 | 2.43E-13 | 4.17E-13 | 2.43E-13 | 2.71E-14 |
| Bupa | 1.69E-14 | 1.69E-14 | 1.69E-14 | 1.69E-14 | 1.69E-14 | 6.14E-14 | 1.69E-14 |
| Car | 8.80E-13 | 6.08E-13 | 6.08E-13 | 5.20E-12 | 3.53E-12 | 8.80E-13 | 8.80E-13 |
| Chart | 4.64E-13 | 3.50E-13 | 1.55E-13 | 5.59E-13 | 6.58E-13 | 5.05E-13 | 4.64E-13 |
| Digits | 2.43E-11 | 2.60E-11 | 9.07E-12 | 2.71E-11 | 2.55E-11 | 9.05E-12 | 2.43E-11 |
| Glass | 8.73E-12 | 7.07E-12 | 7.89E-13 | 1.20E-11 | 1.26E-11 | 1.64E-11 | 8.73E-12 |
| Heart_disease | 8.71E-13 | 1.04E-12 | 8.78E-13 | 9.86E-13 | 8.34E-13 | 1.09E-12 | 8.71E-13 |
| Indian | 1.63E-11 | 7.79E-12 | 1.38E-11 | 1.86E-11 | 2.16E-11 | 2.64E-11 | 1.63E-11 |
| Ionosphere | 9.88E-12 | 1.68E-11 | 1.34E-11 | 1.53E-11 | 1.22E-11 | 1.71E-11 | 9.88E-12 |
| Iris | 1.69E-14 | 1.69E-14 | 1.69E-14 | 1.69E-14 | 1.69E-14 | 1.69E-14 | 1.69E-14 |
| Planning | 1.56E-11 | 1.84E-11 | 9.88E-12 | 1.56E-11 | 1.96E-11 | 8.04E-12 | 1.56E-11 |
| Wine | 1.69E-14 | 1.69E-14 | 1.69E-14 | 1.69E-14 | 1.69E-14 | 6.12E-14 | 1.69E-14 |
| Zoo | 2.71E-14 | 1.18E-13 | 1.69E-14 | 2.43E-13 | 2.90E-13 | 7.29E-13 | 2.71E-14 |
| w\|t\|l | 15\|0\|0 | 15\|0\|0 | 15\|0\|0 | 15\|0\|0 | 15\|0\|0 | 15\|0\|0 | 15\|0\|0 |

realized the selection of the least number of features on 6 datasets, and IWOAIKFS realized the selection of the least number of features on 5 datasets. The proposed method is slightly inferior to the ASO algorithm, but the difference between the two is not much. Therefore, it can be considered that the IWOAIKFS and ASO algorithm have the same advantages in the selection of the minimum number of features. However, if it is analyzed as a whole, the proposed method has better superior performance than the rest of the algorithms in this paper.

In summary, analyzing from a single indicator, despite the effective results in some aspects, IWOAIKFS does not show the best search performance in all indicators, and the algorithm runs longer in experiments, causing certain limitations. Therefore, for IWOAIKFS and other meta-heuristic FS methods, the search and evaluation of the optimal feature subset should be changed according to the data set and actual needs to find the optimal feature subset in a specific scenario.

## Conclusions

In this work, we start from two directions of exploring WOA optimization performance and applying intelligent optimization algorithm to solve the FS, and form the following conclusions through numerical simulation experiments and theoretical analysis.

1. We propose an improved whale optimization algorithm (IWOA). Aiming at the shortcomings of slow optimization speed and low convergence accuracy of WOA, this method first uses chaotic reverse elite individuals to improve the diversity of the initial population of the algorithm. Then, We improve the traditional whale optimization algorithm by simulating the individual preference of whales to hunt prey and the nonlinear weight update

**Table 12.** Friedman test results for datasets.

| *p*-Value | IWOAFS | ASO | GWO | HHO | SCA | SSA | WOA |
|---|---|---|---|---|---|---|---|
| 3.66E-12 | 1.53 | 2.87 | 3.47 | 2.73 | 4.80 | 6.00 | 6.60 |

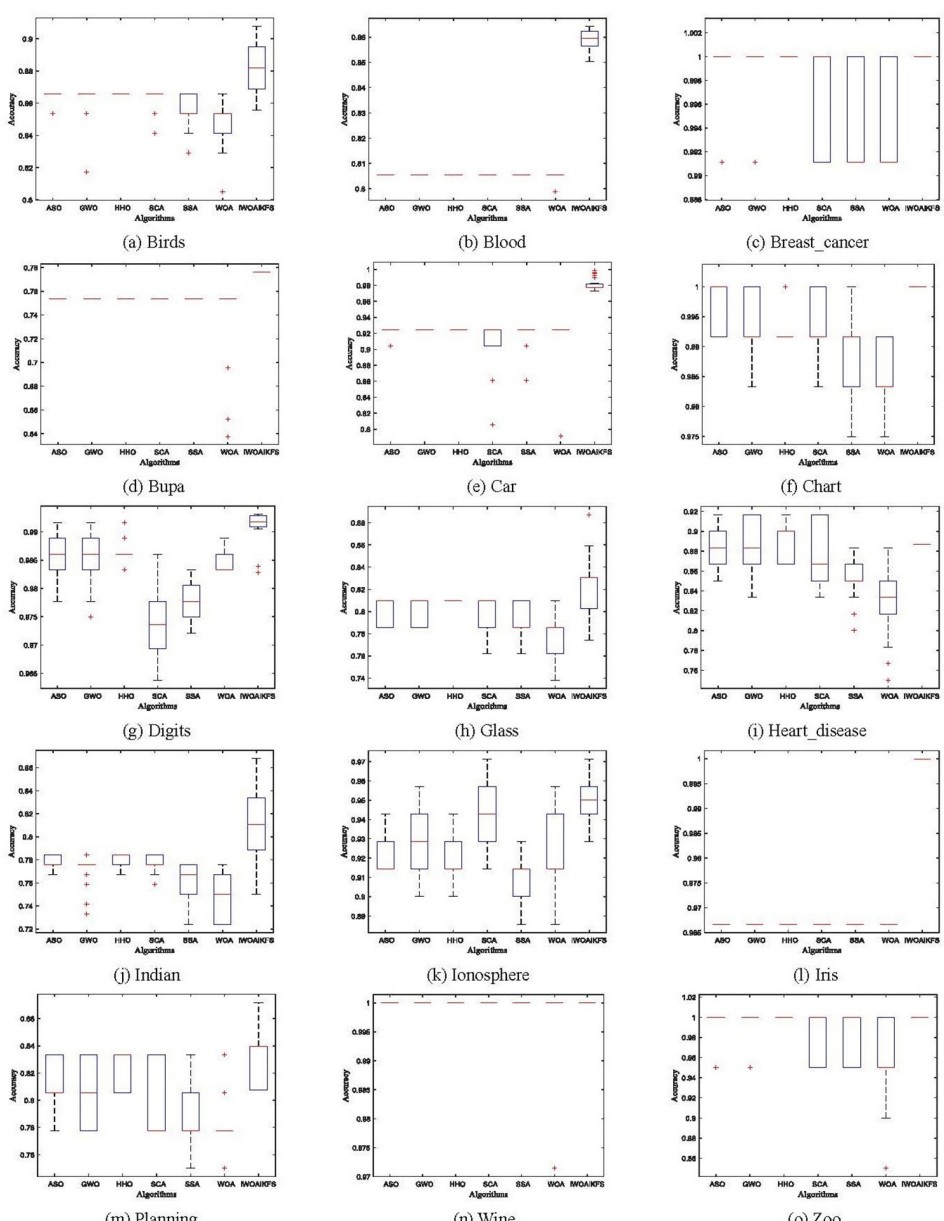

**Fig 12. Boxplot of IWOAIKFS versus other algorithms over all datasets ($k$ = 5).** (a) Birds, (b) Blood, (c) Breast_cancer, (d) Bupa, (e) Car, (f) Chart, (g) Digits, (h) Glass, (i) Heart_disease, (j) Indian, (k) Ionosphere, (l) Iris, (m) Planning, (n) Wine, (o) Zoo.

mechanism of whale group position movement. Finally, the experimental results of 8 benchmark functions and 8 meta-heuristic algorithms in different dimensions show that: compared with the comparison algorithm, IWOA has smaller mean, standard deviation and average fitness in the overall function optimization, and shows better statistically significant performance in Wilcoxon's test and Friedman test, which verifies that IWOA has higher convergence performance and local extremum escape ability.

2. We propose an improved K-Nearest Neighbor algorithm (IKNN). Aiming at the problem that KNN cannot distinguish the similarity between samples by using simple Euclidean

distance as the measure of similarity between samples, this method constructs the similarity measure matrix M between samples through simulated annealing algorithm, and improves KNN by combining with the new weighted classification. The experimental results with 5 classifiers under 8 benchmark UCI datasets show that the higher classification accuracy indicates that IKNN has better classification performance than the comparison algorithm.

3. We propose a feature selection method based on IWOA and IKNN. Aiming at the problem that the traditional feature selection method selects a large number of features and low classification accuracy when screening the optimal feature subset, this method uses the powerful optimization ability of IWOA to search for feature subsets, and uses the powerful classification performance of IKNN to evaluate feature subsets, at the same time, IKNN is optimized synchronously through IWOA, so as to obtain an improved feature selection method. We conduct simulation experiments and analysis on 15 UCI benchmark datasets with IWOAIKFS and 6 optimizers. The experimental results show that on the whole, IWOAIKFS can filter out fewer feature subsets and has higher classification accuracy, showing better search and convergence performance. In addition, the test results of Wilcoxon's test and Friedman test also show that IWOAIKFS has better statistical significance, which further verifies the validity of IWOAIKFS.

Although the three improved methods proposed in this paper have better performance than the original algorithm, they still have some shortcomings. For example, IWOA has poor convergence performance when dealing with high-dimensional multimodal functions, and the time complexity of IKNN and IWOAIKFS is too high. Therefore, we will conduct further research on these issues in the future, as follows.

1. In the future, we plan to build a theoretical analysis system and evaluation system for meta-heuristic algorithms, as well as a community communication module. Due to the problem of over-using "metaphor" in the meta-heuristic algorithm, In order to better distinguish the new meta-heuristic algorithm Whether (or improving the algorithm) can promote the research in the field of optimization. the follow-up research in this paper will try to establish a theoretical analysis system and evaluation system and a community communication module for the corresponding meta-heuristic algorithm.

2. In the future, we plan to try to reduce the time complexity of IWOAIKFS. Since IWOAIKFS is the fusion of IWOA and IKNN algorithm, and influenced by IKNN algorithm, its time complexity is much higher than that of common feature selection methods. Therefore, follow-up research will try to integrate the training and testing processes in IKNN to reduce the time complexity of the IKNN algorithm, thereby reducing the time complexity of IWOAIKFS.

3. In the future, we plan to build a large data set preprocessing system based on IWOAIKFS. After we have built the evaluation framework of the meta-heuristic algorithm and reduced the time complexity of IWOAIKFS, we can try to build a large data set preprocessing system based on IWOAIKFS, which is used to quickly process complex data sets for faster entry into machine learning.

## Supporting information

**S1 Data.**
(ZIP)

## Author Contributions

**Conceptualization:** Zhiqing Guo.

**Data curation:** Zhiqing Guo, Feng Jiang, Zishun Ni.

**Formal analysis:** Feng Jiang, Zishun Ni.

**Funding acquisition:** Guangwei Liu.

**Investigation:** Zhiqing Guo.

**Methodology:** Zhiqing Guo, Zishun Ni.

**Project administration:** Wei Liu, Guangwei Liu, Dong Wang.

**Resources:** Dong Wang, Zishun Ni.

**Supervision:** Wei Liu.

**Validation:** Wei Liu, Zhiqing Guo.

**Visualization:** Wei Liu, Zhiqing Guo.

**Writing – original draft:** Zhiqing Guo.

**Writing – review & editing:** Zhiqing Guo.

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
