## [Decision Letter · Decision Letter 0]

3 Feb 2022

PONE-D-22-01047Improved WOA and Its Application in Feature SelectionPLOS ONE

Dear Dr. GUO,

Thank you for submitting your manuscript to PLOS ONE. After careful consideration, we feel that it has merit but does not fully meet PLOS ONE’s publication criteria as it currently stands. Therefore, we invite you to submit a revised version of the manuscript that addresses the points raised during the review process.

We look forward to receiving your revised manuscript.

Kind regards,

Seyedali Mirjalili

Academic Editor

PLOS ONE

Journal Requirements:

2. PLOS requires an ORCID iD for the corresponding author in Editorial Manager on papers submitted after December 6th, 2016. Please ensure that you have an ORCID iD and that it is validated in Editorial Manager. To do this, go to ‘Update my Information’ (in the upper left-hand corner of the main menu), and click on the Fetch/Validate link next to the ORCID field. This will take you to the ORCID site and allow you to create a new iD or authenticate a pre-existing iD in Editorial Manager. Please see the following video for instructions on linking an ORCID iD to your Editorial Manager account: https://www.youtube.com/watch?v=_xcclfuvtxQ.

Reviewers' comments:

Reviewer's Responses to Questions

**Comments to the Author**

1. Is the manuscript technically sound, and do the data support the conclusions?

Reviewer #1: Yes

Reviewer #2: Yes

2. Has the statistical analysis been performed appropriately and rigorously? 

Reviewer #1: Yes

Reviewer #2: Yes

3. Have the authors made all data underlying the findings in their manuscript fully available?

Reviewer #1: No

Reviewer #2: Yes

4. Is the manuscript presented in an intelligible fashion and written in standard English?

Reviewer #1: Yes

Reviewer #2: Yes

5. Review Comments to the Author

Reviewer #1: This paper presents a method called Improved WOA and Its Application in Feature Selection.

1. The parameters used in all algorithms are given in Table 3. It should be stated why these parameter levels are used/chosen. The performances are highly dependent on the chosen levels and for obtaining best solutions one should determine the optimal parameter set for each algorithm.

2. Would you mind please share the codes to check the performance and data.

3. The English writing should be further polished for readability.

4. The introduction paragraph should be presented more extensively.

5.The paragraph “Conclusions” should be enlarged highlighting the innovative contribution of the paper. Please clarify research in conclusions in modest way. Conclusions can be expanded to get clear understating of major findings.

6. it will be good to provide pros and cons of new proposed method.

Reviewer #2: The manuscript, in its present form, contains several weaknesses. Adequate revisions to the following points should be undertaken to justify the recommendation for publication.

1: The previous work section is petite. I recommend adding new meta-heuristic algorithms to this section as well (Remora optimization algorithm, African vulture optimization algorithm, gorilla troops optimization algorithm, etc.).

2: Please add the system specifications used for the evaluation as well as the programming language.

3: Use Friedman statistical test to better evaluation of a proposed method.

4: What method have you used to transfer solutions from continuous search space to binary space? Describe in full.

5: Please add a new section then explain the computational complexity of a proposed method.

6: What is the value of K in the KNN algorithm? Please explain the main reason for the K-value used.

7: Please add future work to the conclusion section and discuss it briefly.

8: What is the main reason for choosing the Gauss/mouse Chaos map, please elaborate more

9: Use other criteria to evaluate the results

10: All the sections and subsections must be included in the text, Such as :(IWOA comparative experiment, etc.).

6. PLOS authors have the option to publish the peer review history of their article (what does this mean?). If published, this will include your full peer review and any attached files.

Reviewer #1: **Yes: **Nima Khodadadi

Reviewer #2: No

---

## [Author Response · Author response to Decision Letter 0]

22 Mar 2022

Dear reviewers:

Equations and Figures are present in the 'Response to Reviewers', so I cannot be uploaded here.

Therefore, I put all the response in the attachment ('Response to Reviewers').

I have responded item by item to the valuable comments made by the reviewers in the 'Response to Reviewers'

See 'Response to Reviewers' and 'Revised Manuscript with Track Changes' for details

thank you very much!

---

## [Decision Letter · Decision Letter 1]

1 Apr 2022

Improved WOA and Its Application in Feature Selection

PONE-D-22-01047R1

Dear Dr. GUO,

We’re pleased to inform you that your manuscript has been judged scientifically suitable for publication and will be formally accepted for publication once it meets all outstanding technical requirements.

Kind regards,

Seyedali Mirjalili

Academic Editor

PLOS ONE

Additional Editor Comments (optional):

Reviewers' comments:

Reviewer's Responses to Questions

**Comments to the Author**

1. If the authors have adequately addressed your comments raised in a previous round of review and you feel that this manuscript is now acceptable for publication, you may indicate that here to bypass the “Comments to the Author” section, enter your conflict of interest statement in the “Confidential to Editor” section, and submit your "Accept" recommendation.

Reviewer #1: All comments have been addressed

Reviewer #2: All comments have been addressed

2. Is the manuscript technically sound, and do the data support the conclusions?

Reviewer #1: Yes

Reviewer #2: Yes

3. Has the statistical analysis been performed appropriately and rigorously? 

Reviewer #1: Yes

Reviewer #2: Yes

4. Have the authors made all data underlying the findings in their manuscript fully available?

Reviewer #1: Yes

Reviewer #2: Yes

5. Is the manuscript presented in an intelligible fashion and written in standard English?

Reviewer #1: Yes

Reviewer #2: Yes

6. Review Comments to the Author

Reviewer #1: (No Response)

Reviewer #2: The Authors considered all of my comments, So In my opinion this paper will be accepted in this journal.

7. PLOS authors have the option to publish the peer review history of their article (what does this mean?). If published, this will include your full peer review and any attached files.

Reviewer #1: **Yes: **Nima Khodadadi

Reviewer #2: No

---

## [Editor Report · Acceptance letter]

12 Apr 2022

PONE-D-22-01047R1 

Improved WOA and Its Application in Feature Selection 

Dear Dr. Guo:

I'm pleased to inform you that your manuscript has been deemed suitable for publication in PLOS ONE. Congratulations! Your manuscript is now with our production department. 

Kind regards, 

on behalf of

Prof. Seyedali Mirjalili 

Academic Editor

PLOS ONE